# Transcriptomic Studies of Antidepressant Action in Rodent Models of Depression: A First Meta-Analysis

**DOI:** 10.3390/ijms232113543

**Published:** 2022-11-04

**Authors:** El Chérif Ibrahim, Victor Gorgievski, Pilar Ortiz-Teba, Raoul Belzeaux, Gustavo Turecki, Etienne Sibille, Guillaume Charbonnier, Eleni T. Tzavara

**Affiliations:** 1Aix-Marseille Univ, CNRS, INT, Inst Neurosci Timone, 13005 Marseille, France; 2Fondation FondaMental, 94000 Créteil, France; 3Université Paris Cité, CNRS, Integrative Neuroscience and Cognition Center, 75006 Paris, France; 4Sorbonne Université, INSERM, CNRS, Neuroscience Paris Seine, Institut de Biologie Paris Seine, 75005 Paris, France; 5Laboratory of Neuropharmacology, Department of Medicine and Life Sciences, Universitat Pompeu Fabra, 08002 Barcelona, Spain; 6Pôle Universitaire de Psychiatrie, CHU de Montpellier, 34295 Montpellier, France; 7Douglas Mental Health University Institute, McGill University, Montreal, QC H4H 1R3, Canada; 8Department of Psychiatry, University of Toronto, Toronto, ON M5T 1R8, Canada; 9Aix-Marseille Univ, INSERM, TAGC, 13009 Marseille, France; 10Hôpital Sainte Marguerite AP-HM, Pôle de Psychiatrie, 13274 Marseille, France

**Keywords:** major depression, antidepressant, fluoxetine, hippocampus, transcriptomics

## Abstract

Antidepressants (ADs) are, for now, the best everyday treatment we have for moderate to severe major depressive episodes (MDEs). ADs are among the most prescribed drugs in the Western Hemisphere; however, the trial-and-error prescription strategy and side-effects leave a lot to be desired. More than 60% of patients suffering from major depression fail to respond to the first AD they are prescribed. For those who respond, full response is only observed after several weeks of treatment. In addition, there are no biomarkers that could help with therapeutic decisions; meanwhile, this is already true in cancer and other fields of medicine. For years, many investigators have been working to decipher the underlying mechanisms of AD response. Here, we provide the first systematic review of animal models. We thoroughly searched all the studies involving rodents, profiling transcriptomic alterations consecutive to AD treatment in naïve animals or in animals subjected to stress-induced models of depression. We have been confronted by an important heterogeneity regarding the drugs and the experimental settings. Thus, we perform a meta-analysis of the AD signature of fluoxetine (FLX) in the hippocampus, the most studied target. Among genes and pathways consistently modulated across species, we identify both old players of AD action and novel transcriptional biomarker candidates that warrant further investigation. We discuss the most prominent transcripts (immediate early genes and activity-dependent synaptic plasticity pathways). We also stress the need for systematic studies of AD action in animal models that span across sex, peripheral and central tissues, and pharmacological classes.

## 1. Introduction

Major depression, which affects nearly 5% of the world’s adult population, has become the leading cause of disability, and thus represents a heavy burden for our societies [1,2,3]. However, the clinical management of major depression remains “artisanal”, involving many “trial-and-error” attempts based on the intuition of the medical practitioner [4].

Moderate to severe depressive episodes are treated with antidepressant (AD) drugs including selective serotonin reuptake inhibitors (SSRIs), serotonin and norepinephrine reuptake inhibitors (SNRIs), and older, less selective compounds. More than 30 different ADs are available today at an ambulatory setting, and there is no clear difference in overall efficacy between the different AD treatments [5]. Despite their known activity toward targets involved in the pathophysiology of major depression [6] (all of these ADs but the newly approved ketamine target the monoaminergic system, acting on a more or less selective combination of monoamine transporters and/or receptors), they do not provide an effective response for many patients. More than 60% of patients suffering from a major depressive episode (MDE) fail to achieve remission following the AD they are prescribed. Even more, one-third of patients will not reach remission after up to four AD trials [7]. For those who respond, full response is only observed after several weeks of treatment. This delay negatively affects patient compliance, increases overall pharmacological burden, and potentially dangerous side effects.

The discovery of novel targets for better and safer Ads, as well as the identification of biomarkers related to therapeutic efficacy, is urgently needed [8]. For the moment, despite efforts in this direction, there is no reliable biomarker that can predict the therapeutic response in a patient suffering from an MDE, and there is no absolute predictor to guide the choice of the therapeutic approach [9]. For this, we believe it is necessary to know exactly what the effects of AD drugs are, in particular the changes in the gene expression program that they induce over time [10], in the parts of our body directly involved in the processing of emotions and mood.

Indeed, RNA, as an immediate product of gene expression and epigenetic programs, is a perfect reflection of the functional status of an individual. For this reason, many investigators have compared peripheral RNA expression profiles between subjects suffering from major depression and control subjects or between patients at different times during their AD treatment [11,12]. Unfortunately, the complexity of the conditions of human subjects in terms of personal history (stress at different times of life, age, sex, obesity, cardiovascular problems, infectious history, tobacco and alcohol consumption, hereditary factors, and dietary habits) and the different undergoing pharmacological treatments at the time of study require very large cohorts to achieve sufficient statistical power and allow the identification of universal signatures. Furthermore, it would be necessary to study the signatures induced by AD drugs in a context of healthy subjects to compare them to those of subjects suffering from an MDE if we want to be able to effectively select new therapeutic targets. However, from this point of view, the data available in humans are very limited.

To better understand and manage affective disorders, major efforts have been made in recent years to characterize the transcriptional signatures of major depression and AD action that distinguish patients from healthy subjects or their equivalent in animal models, particularly in mice and rats [13]. Animal models provide a relatively homogeneous study population in which the events triggering the onset of depressive-type symptoms can be precisely controlled, as well as the timing of the administration of a particular drug at a specific dose and for a defined duration [14]. In addition, animal models allow access to all tissues that may be relevant for both pathophysiology and biomarker development, which is not always the case in humans for evident ethical and technical reasons. Furthermore, it should be noted that in humans, for the time being, most of the transcriptional data have been obtained from post-mortem brain tissue and that it is complicated in these circumstances to disentangle the signature of treatments from that of the long disease process and the terminal effect of death itself. This is something that can be much better controlled in an animal model where, also, new technologies may rapidly advance our understanding of psychiatric disorders at the cellular, molecular, and brain circuit levels [15].

Much progress has been made on the mechanisms leading to the phenomena of vulnerability or, on the contrary, resilience to stress using animal models and the modalities of the response to AD treatment, in particular with regard to transcriptional signatures [16,17,18,19,20]. As in humans, animal models reveal absence of homogeneous response, which offers the possibility of selecting animals that respond well to AD treatment to study the biological pathways involved in the recovery of normal social, emotional, hedonic, affective, mnemonic, exploratory, and feeding behaviors [21]. Going a step beyond, recent convergence studies have been undertaken from human and animal transcriptomes and should advance our knowledge of the pathophysiology of major depression [16,22,23,24,25]. Concerning the identification of signatures of AD treatments, several teams have tried to gather common features of genome-wide transcriptional variations induced by the action of ADs in patients suffering from affective disorders [11,12,26], but to our knowledge, studies carried out in rodent models are very scarce and fragmented.

Despite the privileged use of SSRIs in the pharmacological armamentarium to treat major depression, and the widespread use of this pharmacological class in omics studies in humans and animal models [27], we found only one review regarding studies conducted in animals to evaluate the transcriptional effects of these drugs [28]. However, this work is necessary to establish the targets common to humans and animals that would deserve special attention to improve therapeutic efficacy, which remains too limited today.

Since, to our knowledge, no study has established common patterns of gene expression program variation in rodent models of AD drug exposure, we searched all available transcriptomic data in rats and mice with AD drug exposure and sought to define recurrent elements of a biological signature with already known biological processes involved in mood disorder pathophysiology, as well as other processes that should merit further investigations.

## 2. Results

### 2.1. Preponderance of Studies Investigating the Effect of Fluoxetine

We first listed all studies reporting effects of a confirmed or putative antidepressant intervention on the variation of transcriptional expression in the mouse or the rat in (i) naïve animals and (ii) in animals subjected to a paradigm modeling negative affects and depression-like behaviors. Focusing on studies using sufficiently mature technologies and considering the whole genome, we identified 108 published articles and 6 publicly available but unpublished transcriptomic datasets (Figure 1, Appendix A). We noticed, however, that in some cases the same animal cohorts have been used in multiple publications. We inventoried 107 independent animal cohorts.

*Sex:* Strikingly, most of the studies were conducted on male animals since out of 107 cohorts only 8 involved females [29,30,31,32,33,34,35,36,37,38], and, in fact, RNA profiling on males and females subjected to the same paradigm was performed only four times [32,33,34,36,37,38]. In contrast, the proportions are much more balanced between studies in mice and rats or between studies conducted in naïve or stressed animals (Figure 2A–C).

*Nature of AD interventions*: When we look at the class of the AD treatments tested (Figure 2D), we see that almost half of the cohorts used SSRIs, in particular fluoxetine (FLX), which was used in more than a third of the studies. The second most represented class is tricyclics, especially imipramine, followed by the NMDA receptor (NMDAR) antagonists, essentially ketamine. Only after that do we find the SNRIs, followed by mood stabilizers (lithium), atypical antipsychotics, electrical stimulation of the brain (essentially by electroconvulsive therapy, ECT), monoamine oxidase inhibitors (MAOIs), and histone deacetylase (HDAC) inhibitors. Finally, while there is a wide variety of AD interventions (both drugs and non-drugs), of the 65 ADs we have listed (Appendix A), more than half (N = 40) have only been tested for pan-genomic activity once.

*Animal models*: Numerous paradigms have been developed in rodents to model symptoms related to human mood disorders and this diversity is reflected in the transcriptomic studies that we surveyed (Figure 2E). Thus, while no fewer than 13 different paradigms have been used (Appendix A), the most popular is unpredictable chronic moderate stress (UCMS) used in 40% of studies, while electric shocks, restraint stress, social defeat, and maternal separation each concerned 8–10% of the studies.

*Region of interest*: From an anatomical point of view (Figure 2F), the brain is the most studied organ, overwhelmingly with 105 out of 107 cohorts, while for the periphery, just three studies profiled blood [13,21,39,40], and only one study examined the adrenal gland [41], liver and kidney [42,43], or mammary glands [31]. Concerning the brain, there is a strong heterogeneity in the size of the cerebral areas or the type of cells profiled ranging from the whole brain to the paraventricular nucleus of the hypothalamus [44] to a particular type of cortical neurons [37,45,46], but, in fact, the most studied area is the hippocampus (as a whole or more precisely the dentate gyri) that we found in almost half of the studies, far ahead of the prefrontal cortex, the frontal cortex, the amygdala and the nucleus accumbens. Finally, although animal models offer the possibility of easily recovering several different anatomical areas and thus determine the possible regionalization of expression profiles, most studies only sampled one anatomical area and only four studies distinguished more than three anatomical areas from the same animals [30,47,48,49].

Given the great variability in terms of AD intervention, paradigms used, species, sex, brain regions, and assay methods, it is difficult to make meaningful comparison across treatments. This is even more difficult considering the fact that a little more than half of the expression data are not publicly available (Figure 1). Therefore, we decided to focus on the effect of FLX on the hippocampus, which nevertheless represents 12 studies (Table 1).

### 2.2. Signature of FLX Response in Stressed Rodents

After retrieving the complete data for each of the 12 selected studies, we established for each dataset the lists of differentially expressed genes between the control condition and the FLX treatment condition. First, we grouped studies in mice and rats where FLX treatment restored behavior in stressed animals, which represents four studies in mice and three in rats (Table 1). We felt that it would not be appropriate to merge all the raw data (after different calibration procedures) into one large dataset because of the great heterogeneity in the specifics of each study. In addition to the differences between species and strains, sample sizes, transcriptomic platforms, and behavioral protocols also differed between studies. Instead, we kept from each individual study the list of the top 300 genes with nominal *p* values below the arbitrary threshold of 0.05. Then, we combined the individual lists into a single graph to allow identification of genes with repeated dysregulated expression in multiple experiments (Appendix A). In this way, we extracted 1973 genes that were modulated by FLX in at least one experiment in stressed animals. Of these genes, 98 were dysregulated in at least four different experiments (Appendix A). We also defined for each gene the global trend of the direction of variation of the transcriptional expression with a positive consensus score for overexpression and negative for underexpression. Of note, although nine genes (*Arc*, *Ddah1*, *Egr1*, *Hmgcs1*, *Kcng2*, *Klhl5*, *Nr4a1*, *Oxtr*, and *Zfhx2*) show significant expression variation in at least five of the experiments, only four show a clear directional pattern (*Ddah1* and *Oxtr* are predominantly upregulated, whereas *Klhl5* and *Zfhx2* are predominantly downregulated). Interestingly, at best, only 15 genes show a significant direction of variation that is consistent across most experiments (Table 2).

In the literature there is no consensus way to establish a signature based on a collection of different datasets. As an alternative approach, we considered defining the FLX action portrait based on the methodology recently proposed by Stephen Gammie to establish a signature of major depression in the human brain [23]. Applying this methodology, we ranked 39,629 genes and obtained scores ranging from +4.56 for the most overexpressed consensually in the seven studies (immediate early response gene, *Ier5*) to −4.46 for the most underexpressed (*Sh3d19*, Appendix A). We listed in Table 3 the signature of effect of FLX with genes showing absolute values of portrait scores equal or above 4. When we compared the integration (Table 2, N = 22) with the portrait score (Table 3, N = 12) signature, we identified two genes (*Hmgcs1* and *Prkar1b*) that are common to both signatures. The advantage of having a sign indicating the overall direction of expression variation, whether with the consensus or the portrait score, allows us to assess whether there is any distortion in the directionality of FLX-induced transcriptional expression variation. We can see that this is not the case because with both scoring methods a little less than 15% of the genes show a directionality of their expression variation in one direction or another and thus for nearly 70% of the genes no trend is apparent. To extract the most salient variations with the portrait score, we separated the genes with scores whose absolute value is equal or greater than 3 and thus obtained 82 overexpressed genes and 80 underexpressed genes (Appendix A). As a more general and consensual signature of FLX in a stress paradigm, we combined both scoring procedures with absolute values equal or above 2, resulting in 412 upregulated genes and 411 downregulated genes in an almost perfect balance (Appendix A).

To identify the biological processes targeted by the signatures described above, we conducted ontological analysis with the DAVID algorithm. Appendix A shows ontological results for the list of the 98 most commonly deregulated genes from Appendix A (integration method). Two overrepresented processes emerged: the activity of transcription factors and the synapse. Then, we performed ontological analysis on the 823 genes with consensus and portrait scores above 2 after separating up- (Table 4) and downregulated genes (Table 5). Overexpressed genes were strongly associated with the synapse, postsynaptic density, and cell junction, as well as with glutamatergic and oxytocin signaling pathways. We also found recurrent modules classically associated with AD function. On the contrary, underexpressed genes were largely enriched for the translation machinery, the ribosome.

### 2.3. Signature of FLX Response in Naive Rodents

Regarding the signature of the effect of FLX on naïve animals, we surveyed seven different datasets, five in mice and two in rats; 1763 genes showed a variation in expression in at least one experiment (Appendix A) and 117 in at least four experiments (Appendix A). Sixteen genes were significantly affected by FLX in five out of seven experiments, and in fact, 36 genes showed significant variation in the same direction in most experiments (Table 6). The most significantly upregulated gene is *Myo1e* (five experiments), whereas the most significantly downregulated genes are *Isoc1*, *Map1a*, *Scn3b*, and *Zfhx2* (five experiments). The application of the portrait scoring method allowed us to rank 36,483 genes with a score ranging from +5.56, for the most overexpressed consensually in the seven studies (vasoactive intestinal peptide, *Vip*), to −5.58 for the most underexpressed (intersectin 1, *Itsn1*, Appendix A). We listed in Table 7 the portrait signature of effect of FLX with 60 genes showing absolute values of portrait scores above 4. Eight upregulated genes (*Cfh*, *Ddr1*, *Gsn*, *Homer1*, *Igfbp6*, *Knstrn*, *Sel1l3*, and *Sema3a*) and nine downregulated genes (*Doc2b*, *Fat4*, *Itga4*, *Itsn1*, *Pcdh19*, *Rasgrf1*, *Scn3b*, *Tnxb*, and *Zfp316*) are common to the signatures we obtained with both methods (Table 6 and Table 7).

The ontological analysis of the processes enriched from the list of 117 genes (Appendix A) reveals again a synaptic location with a glutamatergic signaling (Appendix A). When performing ontological analysis on the 528 upregulated genes with consensus and portrait scores above or equal to 2, no real specific process was significantly enriched (Appendix A). For the 613 downregulated genes, cell junction and synapse were the most significant locations and regulation of NMDA receptor activity, insulin secretion, and circadian entrainment were also affected (Table 8).

### 2.4. Shared Signature of FLX in Stressed and Naïve Rodents

To establish the core of FLX-induced transcriptional activity in the hippocampus, regardless of the paradigm employed, we first examined the convergence between the stress and non-stress signatures previously described and obtained by either integration or scoring methods. Figure 3 shows that only one gene is common to all comparisons, *Zfhx2*. Next, we combined in the same dataset, the 14 comparisons made previously (7 in a stress context and 7 in a naïve context, Appendix A). With the integration method, among 3360 genes modulated in at least one experiment following FLX treatment, 78 were altered in at least 7 experiments, 15 in at least 8 experiments, while *Pdlim5* and *Zfhx2* were altered in 9 and 10 comparisons, respectively (Appendix A). Second, by applying the portrait method to the 14 comparisons, we could rank 40,113 genes with top scorer genes, *Sel1l3* (+7.88) and *Nfia* (−8.13) (Appendix A). When we examine the best scoring genes, with absolute values of both integration and portrait scores equal or above 6, we obtain nine upregulated genes: *Ddr1*, *Ier5*, *Igfbp6*, *Nptx2*, *Prkar1b*, *Ptpn5*, *Sel1l3*, *Tyro3*, and *Zfp703*, and seven downregulated genes: *Akt3*, *Fat4*, *Nfia*, *Pcdh19*, *Rab27a*, *Scn3b*, and *Zfhx2*.

By evaluating lists of equal size obtained by integration method (Appendix A) versus consensus and portrait score (absolute values equal or above 5, Appendix A), we defined 15 FLX-modulated genes: *Baalc*, *Igfbp6*, *Itga4*, *Nptx2*, *Prkar1b*, *Rasgrf1*, *S100a6*, *Sel1l3*, *Slc4a4*, *Sorcs1*, *Tmem47*, *Trpm3*, *Zfhx2*, and *Zfp316*. Among these genes, five are also listed in Figure 3 (*Baalc*, *Igfbp6*, *Prkar1b*, *Sel1l3*, and *Zfhx2*).

Without surprise, the ontological analysis of the processes enriched from the list of 78 genes most affected by FLX action (Appendix A integration method) reveals mainly a synaptic activity (Appendix A). When performing ontological analysis on the 113 upregulated genes with consensus and portrait scores above or equal to 4, the axon and the MAPK signaling pathways were significantly enriched (Table 9). For the 151 downregulated genes, the MAPK signaling pathway was only process significantly affected (Table 10). A secondary analysis of the genes involved in the MAPK signaling pathway identified distinct components for up and downregulated genes. Upregulated genes were associated with neurotrophic pathways while downregulated genes were restricted to protein phosphorylation.

The ontological analysis of the processes enriched from the list of 78 genes deregulated after FLX action confirms with both ontological tools that neuronal structure, development, and signaling are the main processes concerned (Table 8 and Table 9).

## 3. Discussion

In this study we have voluntarily limited our field of research to post-2006 publications on transcriptional signatures of the effect of AD treatments in animal models and have identified 114 studies. Nevertheless, we were confronted with the difficulty of accessing complete transcriptional data for more than 60 studies for which data were not available in a public manner. This unfortunately reflects, at the minimum, a lack of reflexes on the part of investigators, or perhaps also a lack of sufficient incentive on the part of the authors and publishers, to share data [59]. Indeed, omics studies come at a substantial cost, and it is conceivable that some authors would prefer to fully capitalize on future analyses before sharing the data publicly.

### 3.1. Methodological Constraints

#### 3.1.1. Current Picture: A Fragmented Landscape

A detailed review of the available studies shows a fragmented landscape regarding the design of the studies and an important imbalance in many experimental factors regarding sex, region, and/or ADs studied (Figure 1).

Although women are twice as affected by major depression, preclinical study models of this disease and its treatment are still too often focused on male animals. Thus, among all the studies studying the transcriptome following the action of AD molecules, less than 10% concern females (Figure 1). In addition, recent studies suggest that the transcriptional signatures of depression are different in men and women [60]. It is therefore essential that more preclinical studies seriously address the changes induced in females by the major therapeutic agents that are or will be offered to patients with mood disorders.

A similar imbalance exists for brain regions targeted in transcriptomic studies of AD effects (Figure 1). This is an important caveat because, while stress (acute and prolonged) causes molecular and functional changes in several brain areas such as the medial prefrontal cortex, amygdala, nucleus accumbens, and hippocampus, among the most studied, the stress-induced molecular signatures differ across regions. For instance, activation of the CREB-BDNF axis in the hippocampus is antidepressant, but the same activation in the nucleus accumbens is pro-depressant [61]. Region-dependent effects of stress on neuronal activation and BDNF signaling could also be sex-dependent [62,63]. Comparing AD transcriptomic signatures across regions in male and female mice could provide potential insights into how brain pharmacology can be modulated in a region-dependent and sex-specific manner; however, we could not undertake this analysis because of insufficient power.

Another big discrepancy that we observed across studies concerns pharmacological classes tested and compounds inside the same class. Although SSRIs remain the most prescribed ADs, it is important to revisit older, less selective classes of ADs. Indeed, many open questions remain in terms of both the delay of onset [64] and of the mechanism of action [65]. Again, there was not enough power (numbers) in the available studies to consider for our analysis. Therefore, our review of the literature underlines the need for systematic studies designed to include male and female mice, different regions, and more than one class of ADs.

#### 3.1.2. Methodological Choices

It is not a surprise that the majority of the available studies concerns the most prescribed AD in the world, Prozac, approved by the Food and Drug Administration (FDA) in 1987, whose active ingredient is FLX [66]. Moreover, the preferred study area for the transcriptional effect induced by ADs is the hippocampus, a region consistently implicated in major depression and AD action [67]. Thus, we decided to focus on the transcriptomic signature produced by FLX in the hippocampus. In the different studies analyzed for some is the whole hippocampus that has been collected; in other cases, it is the dentate gyrus and/or the cornu ammonis 1 (CA1) and sometimes it is specified whether it is the ventral or dorsal part (Table 1). The hippocampus is functionally divided into a dorsal region that is primarily engaged in cognitive functions and a ventral region that regulates emotion [68,69]. Although it has been observed for one of the experiments we included that similar gene expression responses to FLX were found in dorsal and ventral dentate gyrus [51], more recent investigations show that the ventral hippocampus was particularly sensitive to the effects of stress. Therefore, it was proposed to consider the dorsal and ventral hippocampus separately when conducting high-throughput molecular analyses [24,70]. We are aware of this caveat, but we had no choice if we wanted to keep enough power in our analysis. Stephen Gammie has recently drawn a picture of depression based on publicly available transcriptomic data from human brains [23]. His comparison of the human signature with the signatures of nearly 200 data sets, mainly from mice or rats treated with ADs and other drug classes, showed that the effect of FLX assessed at the hippocampus level provided the closest inverted signature to that of human depression [23].

Another choice we made, for the sake of homogeneity, was to select, at least for one experiment, samples from mice responding to FLX (GSE84185). Indeed, if it seems that most investigators retained for transcriptomic analysis only samples from animals showing an improvement in behavior following FLX administration, in at least one case the overall response was profiled [21]. It may be noted that at least two studies had highlighted interindividual variability in behavioral responses to FLX [21,51]. It was even proposed that it would be interesting to evaluate ambiguous responders, as they could be useful in dissociating anti-depression-like effects from anti-anxiety-like effects [51]. Later, such behavioral heterogeneity, especially concerning anxiety, following chronic FLX treatment was confirmed and may reflect a specific gene expression profile [46].

A limitation of our work is that we almost only processed protein-coding mRNAs. However, emerging literature, especially in terms of biomarker identification for mood disorders, concerns microRNAs (miRNAs) [71,72]. Thus, among the studies that we identified as potentially relevant for our analysis, nine concerned miRNAs, two of which were interested in the effect of FLX in the hippocampus [38,73]. It will therefore be important in the future to compare miRNA and mRNA signatures to study whether they are related [74,75], which will help to better understand the mechanism of action of ADs and undoubtedly improve their effectiveness [76]. In addition to miRNAs, a large family of non-coding RNAs could not be considered in our analysis. These are the myriad of long non-coding RNAs (lncRNAs) [77]. However, not only are they very present for a while on microarrays, and a fortiori are widely detected by high-throughput sequencing, but it is often in this class of RNA that we find the strongest expression variations [78,79]. Efforts to standardize annotations and a better knowledge of their exact role should in the future help us to better integrate them into a more global signature of the effect of ADs [80,81]. Circular RNAs are another category that is nowadays emerging in the field of psychiatry transcriptomics [82]. However, the available data in animal models of depression and antidepressant action are still limited, albeit promising [83].

### 3.2. Biological Pathways and Genes

#### 3.2.1. Serotonin System and BDNF

The primary target of FLX is the serotonin transporter 5-HTT, encoded by the *SLC6A4* gene. Although there may be transient effects on variations in Slc6a4 mRNA expression levels in rodent models [84,85], there is little evidence to suggest a transcriptional effect of FLX on either *SLC6A4* [86] or the limiting enzyme for serotonin synthesis in brain, tryptophan hydroxylase 2 (TPH2) [28], and we did not observe robust modifications for these two genes. On the other hand, in the large family of serotonin receptors, several studies have implicated transcriptional variations in the gene encoding the 1B receptor (*HTR1B*). We confirm in our meta-analysis this effect for *Htr1b* but also for *Htr5b* (Table 6), for which there is no functional equivalent in humans [87]. One target related to serotonin is the overexpressed *Vip* messenger RNA (Table 7, Appendix A) encoding vasoactive intestinal peptide, as it has been known for a long time that the neurotransmitter effect of serotonin in the hippocampus can be modulated by *Vip* through regulation of cyclic adenosine 3’:5’-monophosphate (cAMP) levels and serotonin receptors [88,89,90]. It was also found that FLX, while improving depressed behavior in a rat model of chronic stress-induced depression, increased *Vip* expression [91]. Other expected targets for FLX include brain-derived neurotrophic factor (BDNF), which is widely implicated in depression and the mechanisms of its treatment and would be expected to be increased [28], which is confirmed both in a stress paradigm (Table 4 and Appendix A), or in naive animals (Appendix A) and thus also in the overall meta-analysis (Table 9 and Appendix A).

#### 3.2.2. Immediate Early Genes

In the present meta-analysis, among the genes significantly modulated by FLX appear the immediate early genes *Nr4a1*, *Nr4a2*, *Arc*, *Egr1*, *Fosb*, *Fosl2*, and *Junb* (Table 2, Appendix A). These are genes that have long been identified as particularly important in the program of gene expression modification during the induction of depression in different animal models [92,93,94]. EGR1, in particular, is the major downstream partner of the ERK/Elk-1 cascade that we have recently proposed as a novel target for AD development [67]. Interestingly, such immediate early genes are regulated not only in the hippocampus but also in other brain areas such as frontal and prefrontal cortex, lateral amygdala, and nucleus accumbens [95,96,97,98,99], and they are the indirect target of other AD drugs than FLX such as agomelatine, duloxetine, imipramine, ketamine, paroxetine, or vortioxetine [100,101,102,103,104,105] but also of other AD-related interventions such as sleep deprivation and deep brain stimulation [106,107,108]. In addition to this set of genes, we also noted the immediate early response 5 gene (*Ier5*), which appears upregulated in our meta-analysis (Table 3, Appendix A), was previously found dysregulated in peripheral blood mononuclear cells of unmedicated mood-disorder patients compared to healthy controls [109].

#### 3.2.3. Signal Transduction Pathways

It is not a surprise that our analysis highlights universal signal transduction pathways. Following receptor activation, second messengers fine-tune neuronal activity, synaptic remodeling, long-term potentiation, and neurogenesis. They thus regulate synaptic plasticity and cellular resilience, processes by which the brain perceives, adapts, and responds to a variety of internal and external stimuli (including stress and other depressogenic factors [110]. Over the last decades, evidence from different research groups showed that kinase–phosphatase pathways are important mediators of AD action, and have used translational models, namely genetically modified animals to phenocopy AD-sensitive behaviors and to highlight AD-sensitive brain circuits [111,112,113,114].

aMAP kinases

In line with the neurotrophic hypothesis of depression, genes coding for members of the neurotrophic tyrosine receptor kinase family such as *Ntrk2* and *Ntrk3* were normalized by FLX action (Table 4, Table 6, Table 9 and Appendix A) in a reverse direction compared to observation in rodent stress models of depression [115]. These genes encode kinases that, upon neurotrophin binding, phosphorylate members of the MAPK pathway. Thus, it makes sense to find the MAPK signaling pathway as one of the pathways most consistently affected by FLX (Table 9 and Table 10). Among the multiple connections between BDNF, MAP kinases, and AD action, we found an upregulation of the *Ptpn5* gene expression (Appendix A and Table 9 and Figure 3). It encodes a striatal-enriched tyrosine protein phosphatase (STEP) that inactivates key neuronal signaling proteins such as MAP kinases, tyrosine kinases NMDA, and AMPA receptors and whose inactivation decreases the expression of BDNF [116]. Therefore, STEP inhibitors have been proposed as a novel target for AD drugs of the new generation [117].

bWNT/catenin

Another potential player in this activation of the MAPK pathway is insulin-like growth factor binding protein 6 encoded by *Igfbp6* mRNA, which is overexpressed in the different paradigms in response to FLX (Table 6, Table 7 and Table 9 and Figure 3) including studies not included in the current meta-analysis [118]. There is even another member of this gene family with *Igfbp4* (Figure 3), which can be linked to the activation of another biological pathway important for the resolution of depressive symptoms in connection with hippocampal neurogenesis, the Wnt/β-catenin pathway [119,120]. That same pathway is also activated by the neuronal pentraxin 2 [121], encoded by the *Nptx2* gene that is upregulated by FLX (Table 3, Table 9, Appendix A) as already demonstrated in a previous study focused on a shared mechanism of AD effect of chronic FLX and exercise [118].

cPI3K/AKT

The response to ADs involves several overlapping mechanisms and, in addition to the signaling mentioned above, the PI3K/AKT pathway can also be mentioned [122]. Thus, a dysregulation of *Akt3* expression is observed (Table 7 and Table 10), knowing that the invalidation of this gene in mice revealed an endophenotype reminiscent of psychiatric manifestations such as schizophrenia, anxiety, and depression [123]. The gene *Tyro3*, which we found upregulated after FLX action (Table 4, Table 7, Table 9 and Appendix A), encodes the most widely expressed receptor protein tyrosine kinase and is linked to the PI3K/AKT pathway [124].

dCyclic AMP

It is noteworthy that among the genes common to the signatures of FLX activity in a stress or naive context is the *Pde4b* gene (Table 6 and Appendix A, Figure 3). It encodes a cyclic phosphodiesterase, regulating concentrations of cyclic nucleotides, and thereby plays a role in signal transduction. Not only have several studies specifically shown variations in the expression of transcripts and protein encoded by *Pde4b* in preclinical models and humans [125,126,127,128,129], but this gene is in fact the target of a drug with AD properties, rolipram. A link between neurotrophin synthesis and the production of second messengers like cAMP is protein kinase A. It turns out that one of the genes most reliably affected by FLX codes for a subunit of this kinase, *Prkar1b* (Table 2, Table 3, Table 4 and Table 9 and Figure 3).

eGlucorticoïds

Perhaps a pivot of the effect of antidepressants on major depression through several previously mentioned signaling pathways, cAMP/PKA, phosphodiesterase, and neurotrophins, materializes through the activation of nuclear receptors, the glucocorticoid receptor (GR) and the mineralocorticoid (MR) [130,131,132,133,134]. Two genes share the task of regulating, among others, inflammatory responses, proliferation and differentiation processes, *NR3C1* and *NR3C2*. In our data, if it seems that the tendency is rather to underexpress these two transcription factors, *Nr3c2* is more consensually deregulated in the different signatures that we obtain (Appendix A), without being one of the key markers either. On the other hand, considering the partners of the GR, we notice that AHI1, which regulates the nuclear translocation of the GR, and which has already been associated with stress-induced depressive behavior in mice [135], has its mRNA significantly underexpressed in the different signatures we have generated (Appendix A).

fSynaptic plasticity

In a way, in fine, the beneficial action of SSRIs can be linked to a capacity to achieve synaptic plasticity and from this point of view it seems important to us to mention the decrease in expression of two mRNAs coding for transcription factors involved in brain development [136], *Nfia* and *Nfib* (Table 2, Table 7, Appendix A and Figure 3). For the latter, a study in rats had shown that its expression variation was during the molecular signature at the level of the frontal cortex characterizing the AD action of quetiapine in a chronic stress model in rats [137]. Recently, a study on lymphoblastoid cell lines from depression patients treated with citalopram reported a significant association of *NFIB* expression with improvement in depression scale [138]. The NMDA subtype of glutamate receptors (NMDAR) plays a key role in synaptic plasticity in the context of depression [139], and it has been shown that a Ca^2+^/calmodulin-dependent Ras-guanine-nucleotide-releasing factor (RasGRF1) served as an NMDAR-dependent regulator of the ERK kinase pathway. In our meta-analysis, we noted that *Rasgrf1* was one of the most consistent downregulated genes by FLX activity (Table 6, Table 7, Table 8 and Table 10). Consistently, it has been shown that *Rasgrf1* knockdown reversed the effect of UCMS on mice [140], and it has been shown in humans that RASGRF1 may be a potential specific biomarker of treatment response for bipolar disorder [141]. Concerning the NMDAR-positive modulation underlying AD action, one identified initial trigger of such an effect is Drd1-pyramidal cell signaling [142,143]. We uncovered a *Drd1* gene upregulation induced by FLX in our meta-analysis (Table 4, Table 7 and Table 9) in agreement with these observations. Another gene common to the different signatures of FLX activity is *Pdlim5* (Table 6, Figure 3), which encodes a protein that tethers kinases to the Z-discs in striated muscles but also restrains the postsynaptic growth of excitatory synapses. Several investigations suggested that the human gene plays a key role in the pathophysiology of mood disorders but is also likely involved in AD response [144,145,146].

Overall, whatever the paradigm explored (stress vs. naïve), it appears that FLX has a marked effect on synaptic activity in the hippocampus (Table 4, Table 8, Appendix A). Thus, many genes characterizing the transcriptional signature of FLX code for proteins playing an important role at the synaptic level. This is the case for the *Sorcs1* gene [147], encoding the sortilin-related VPS10 domain containing receptor 1, upregulated by FLX (Table 6 and Appendix A) as previously observed [148]. Another example is the *Itsn1* gene (Table 6, Table 7, Table 8 and Appendix A), encoding the intersectin 1 that regulates synaptic vesicle recycling [149]. Interestingly, we can even point out a link, perhaps not the most expected, between synapse and transcriptional activators. For example, the *Pcdh19* gene (Table 6, Table 7, Appendix A), which codes for a protocadherin, bridges neuronal activity with gene expression by regulating immediate-early genes expression to favor maintenance of neuronal homeostasis [150]. In addition, *Sox11* (Table 7, Appendix A) encodes a transcriptional factor playing a major role in brain development and activated in an activity-dependent fashion specific to the dentate gyrus of the hippocampus [151], which had already been shown to be dysregulated by FLX [118], as well as by paroxetine, another SSRI [100].

#### 3.2.4. Emerging Pathways

A novelty in this meta-analysis is the zinc finger homeobox 2 gene, *Zfhx2* (Table 2 and Table 6, Figure 3), for which there is generally very little literature. This putative transcriptional regulator is highly expressed in the developing brain and, very interestingly, its global knockout has been reported to show several behavioral abnormalities, namely hyperactivity, enhanced depression-like behaviors, and an aberrantly altered anxiety-like phenotype [152], as well as hyposensitivity to noxious mechanical stimuli [153]. A human variant of *ZFHX2* has been associated with a pain-insensitive phenotype [153]. Another gene related to pain and nociception is *Trpm3* (Table 7 and Appendix A, Figure 3), encoding a tetrameric cation channel. It had been shown that FLX but also imipramine and the antipsychotic drug chlorpromazine inhibit the channel activity [154]. Among the genes whose relevance in terms of belonging to a biological pathway is not at first obvious to relate to the pathophysiology of depression or the action of ADs is *Slc4a4* (Table 7, Appendix A). It encodes a sodium bicarbonate cotransporter (NBC) involved in the regulation of bicarbonate secretion and absorption and intracellular pH, but, interestingly, it has been proposed as one of the best biomarkers for predicting suicidal ideation [155,156]. Conversely, our analysis pinpoints to the oxytocin pathway altered in animals under stress exposure (Table 3) that has received renewed attention in regard to psychiatric disorders [157].

Last but not least, components of the ribosome that we saw as particularly affected in stressed animals responding to FLX (Table 5) are also affected in the study of Hori et al., who explored the blood transcriptome of three homogeneous groups, namely “resilient”, “vulnerable”, and “resistant” [158]. Remarkably, among genes dysregulated between the different groups are *RPL17*, *RPL21*, *RPL34*, *RPS15A*, and *RPS27*, which were all dysregulated in rodent models (Table 5). In another study, Guilloux et al. were able to identify a group of six genes with predictive value for response to citalopram treatment for major depression [159]. Half of the genes were ribosomal units including again *RPL17*, as well as *RPL24*, that we found dysregulated (Table 5).

## 4. Methods and Materials 

### 4.1. Systematic Search of the Literature

A systematic search was conducted on PubMed on 19 April 2022. The search was limited to papers published after 2006 to exclude studies based on immature DNA microarrays technology. It used combinations of the following terms: (antidepressant OR fluoxetine) AND (microarray OR transcriptome data OR transcriptomic changes OR RNA-sequencing OR transcriptome profiling OR differential molecular signature OR large-scale gene expression OR bacTRAP) AND (mouse OR rat OR rodent OR depression-like) NOT (focused microarray OR cell line OR cancer OR tumor OR traditional medicine) NOT (review [publication type]). The filter “English language” was also activated. In addition, references from eligible studies were examined to include additional studies that would not be retrieved by the PubMed search. To compile a complete list of eligible studies (Appendix A), we excluded the studies that did not use an AD or profile the action of an AD genome-wide. Because we focused on the in vivo action of AD in rodents as a model of major depression in humans, we also excluded studies performed in systems other than rodent. Namely, studies on other species, cells in culture, and on bacterial infection or brain injury models were excluded. We searched all eligible studies for either an ID number linked to the public download of full transcriptomic data or a table containing the complete data as Appendix A. We also searched NCBI resources, the Gene Expression Omnibus (GEO, https://www.ncbi.nlm.nih.gov/geo/, accessed on 19 April 2022), the Sequence Read Archive (SRA, https://www.ncbi.nlm.nih.gov/sra, accessed on 19 April 2022), and the European repository ArrayExpress (https://www.ebi.ac.uk/arrayexpress/, accessed on 19 April 2022) for microarray and high-throughput sequencing data related to published and unpublished studies by typing “antidepressant AND (mouse OR rat)”. Finally, when full transcriptomic data were not publicly available, we emailed the corresponding authors to request raw data.

### 4.2. Bioinformatic Analyses

The analyses were produced using Snakemake (https://doi.org/10.12688/f1000research.29032.1, accessed on 19 April 2022) as workflow manager and R Bookdown for reporting.

#### 4.2.1. Individual Reprocessing of Published Experiments—Microarrays

Normalized microarrays from seven studies (GSE84185, GSE43261, GSE56028, GSE118669, GSE54307, GSE6476, and GSE42940) were retrieved from GEO using GEOquery [160]. One last microarray was retrieved directly from the authors [50] and was normalized using RMA from the oligo R package [161]. The microarrays’ latest annotations were retrieved from Ensembl Biomart for Affymetrix GeneChip Mouse Genome 430 2.0, Affymetrix GeneChip Mouse 430A 2.0, Affymetrix Rat Gene 1.0 ST arrays, and Agilent SurePrint G3 Mouse Gene Expression 8 × 60 K microarrays. When unavailable, annotations from the original authors were completed using megablast queries (GSE42940). Probes with mapping to multiple genes were discarded. Differential gene expression analyses were produced with Limma [162].

#### 4.2.2. Individual Reprocessing of Published Experiments—RNA-Seq

Raw reads from four studies (SRP056481, SRP057486, SRP084288, and SRP131063) were retrieved from SRA using SRA tools. Reads were trimmed using Sickle (-q 20) [163] and aligned on GRCm38 (for mouse samples) or Rnor6 (for rat samples) with STAR [164] using default settings. Gene-level counts were produced using subread featureCounts [165] with Ensembl release 102 annotations. Differential gene expression analyses were produced with Deseq2 [166] using apeglm shrinkage [167].

### 4.3. Integration Analyses

To compare differential expression analyses between different experiments, we kept only genes with *p* values below 0.05 and in the top 300 of any compared designs. For RNA-Seq datasets, the log2 fold change values have been calculated using the “normal” shrinkage type and not the apeglm from Deseq2 because the latter was considered too stringent for this kind of integration analysis. Plots were produced using the R ggplot2 package [168]. To be able to specify the general trend of the direction of expression variation of a gene, we were inspired by the procedure defined in the MetaVolcano tool, the vote counting approach [169]. This consists of counting the number of comparisons with *p* values below 0.05 showing overexpression and subtracting the number of comparisons with *p* values below 0.05 corresponding to underexpression. The resulting integer, whether positive, zero, or negative, is called the consensus score.

### 4.4. Portraits of FLX Action

As an alternative procedure to extract the transcriptional signature of FLX treatment, we applied the method proposed by Stephen Gammie to generate a portrait of depression [23]. The *p* value for each gene and for each individual microarray or RNA-Seq dataset used in integration analyses was −log10 transformed and then multiplied by the sign of the direction of change. Then, for each dataset, both upregulated and downregulated genes were ranked from low to high. With the 1000-increment cutoff, for each gene we counted how many times it received a ranking of 1000 or less among the upregulated genes. The same approach was applied to count 2000 or less, but that time the count number was multiplied by 0.1. We used this method up to 8000 by multiplying the sum by 10^−(k−1)^, where k is the number of the thousands of ranks considered. The same calculation was applied to downregulated genes. Finally, we calculated the sum of the above values for a given gene. Genes were sorted by the absolute value so that the final list contains all genes ordered by magnitude of the difference from controls with also information on direction of expression change.

### 4.5. Ontological Analysis

Gene lists were uploaded on DAVID (database for annotation, visualization and integrated discovery) Bioinformatics Resources (2021 Update) [170] for identifying statistically relevant biological processes. The default settings were kept unchanged and only functional annotations related to the gene ontology (GO) biological process, molecular process, and cellular component, as well as the KEGG pathway, with corrected *p*-value (Bonferroni) < 0.05, were retrieved. We limited the search to *Mus musculus* species annotation. To simplify the lists of biological processes obtained, we only kept, for identical gene lists, the process associated with the lowest *p*-value.

## 5. Conclusions

We provided the first meta-analysis of a transcriptional signature of FLX in animal models and highlighted the impact and pitfalls of gene expression variation studies. Future studies aiming at extending such signature to other widely prescribed ADs and considering other tissues and brain regions in both males and females are warranted to provide better insights into optimal targets for AD response in human.

Despite all the limitations mentioned before, our study shows how murine models are a valuable source for understanding the mechanisms involved in the response to antidepressants. Indeed, with the benefit of hindsight on the main processes regulated by FLX, we realize that there are solid points of convergence with what is described in the context of the follow-up of patients suffering from major depression. It would obviously be important to know whether the changes induced by FLX, in the context of an animal model recapitulating the response to AD treatment, are similar to the biological program that is set up in a depressed patient who responds favorably to AD treatment. The major concern is that all human studies based on post-mortem tissues cannot distinguish the death-induced signature from those induced by the treatment and those corresponding to the depressive illness. In contrast, studies that allow for clinical interviews at the same time the biological samples are taken, such as studies based on blood samples, can define the AD treatment-related signature in relation to response or non-response to treatment. In fact, studies on human cohorts have already underlined the importance of some biological processes significantly enriched from ontological analyses of genome-wide transcriptional variations in FLX-treated rodent models of major depression. That includes synaptic plasticity [171], even when the variations in the corticolimbic regions of the brain are of opposite directions between men and women [60,172], and in particular at the blood level with glutamatergic signaling, the intervention of neurotrophin [173,174], and MAPK signaling [74].

Overall, our results highlight activity-dependent gene transcription as the major AD target. This is a major finding with an important potential for future biomarker clinical research. At the center of cellular plasticity and resilience, in the CNS and the periphery, such cell signaling cascades are likely early-stage mediators of AD response body-wide. The findings of this meta-analysis allow us to therefore hypothesize that ADs do induce early changes in cellular biology and processes at the interface of CNS and periphery that could constitute clinically relevant biomarkers and suggest that more focus is needed in the earliest point of biomarker variations.

## Figures and Tables

**Figure 1 ijms-23-13543-f001:**
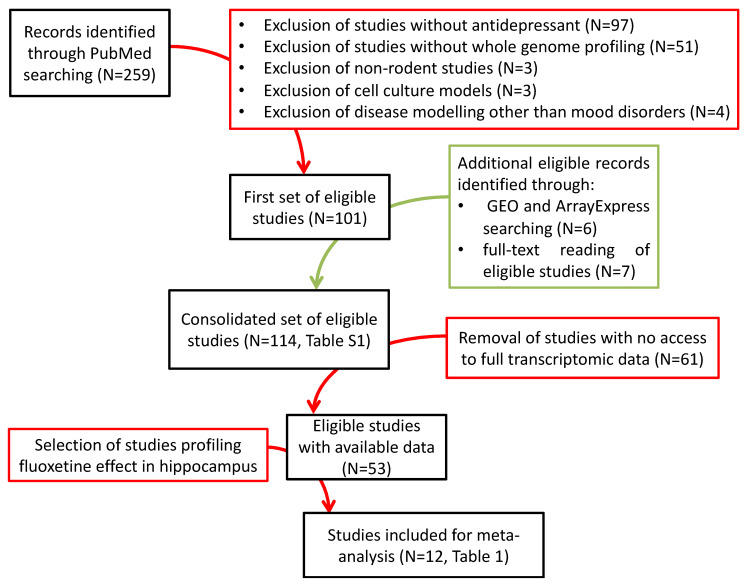
Flow diagram outlining the selection procedure of the studies kept in the meta-analysis.

**Figure 2 ijms-23-13543-f002:**
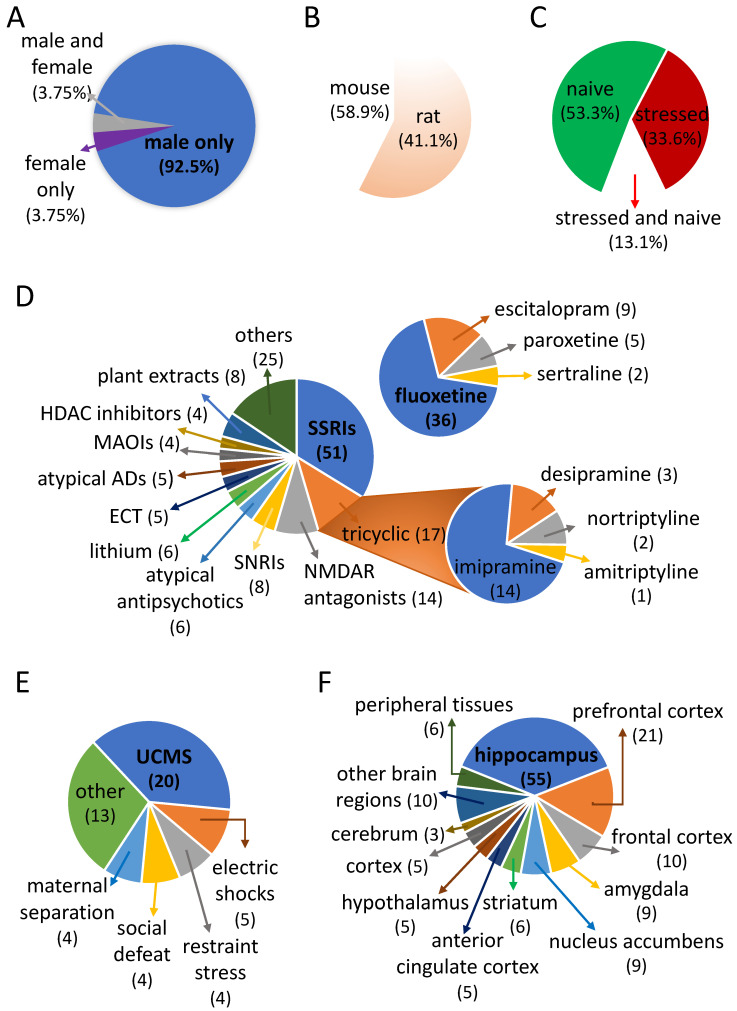
Main features of transcriptomic studies conducted in rodents to investigate the effect of AD treatments. Distribution of the sex (**A**), rodent species (**B**), and use of a stimulation to induce depression-like behaviors before testing the effect of ADs (**C**). (**D**) Diversity of the class of ADs used with detailed proportions for SSRIs and tricyclics. (**E**) Variety of paradigms used to induce depression-like symptoms. (**F**) Anatomical distribution of collected tissues. Numbers in parentheses indicate the number of studies where an AD (**D**), a type of stress (**E**), or a specific piece of tissue (**F**) has been examined.

**Figure 3 ijms-23-13543-f003:**
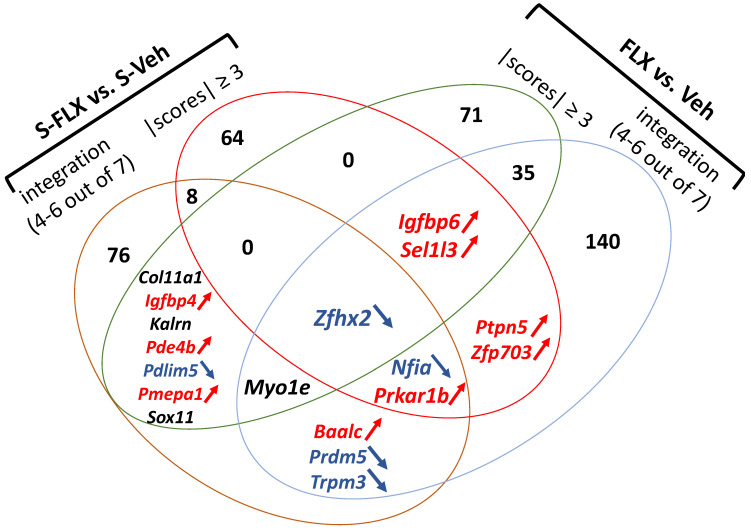
Venn diagram of hippocampal transcriptional signatures. The signatures were induced by either FLX in a stress-inducing depressive-like behavior paradigm (S-FLX vs. S-Veh) or FLX in naïve rodents (FLX vs. Veh) and the list of genes were obtained by either integration (significant in at least four experiments) or scoring methods (both consensus and portrait). Genes in red are generally overexpressed, genes in blue are downregulated whereas genes in black show no consensual trend of variation direction.

**Table 1 ijms-23-13543-t001:** Studies selected for meta-analysis.

Meta-Analysis	ID	Ref.	Rodent	Stress	Fluoxetine	Hippocampus Dissection	Sample Number	Platform
mouse S-FLX vs. S-Veh	GSE84185	[21]	8-week BALB/c	8-week UCMS	In drinking water (10–20 mg/kg/day) for the last 6 weeks	dentate gyrus	N = 21: 8 NS-Veh; 8 S-Veh; 5 S-FLX-R	Agilent SurePrint G3 Mouse Gene Expression 8 × 60 K
	private	[50]	8-week BALB/c	7-week UCMS	ip administration (20 mg/kg/day) from week 2 to the end of UCMS	dentate gyrus 5 h after the last injection	N = 18: 6 NS-Sal; 6 S-Sal; 6 S-FLX	Affymetrix GeneChip Mouse Genome 430 2.0
	SRP057486	[48]	7–8-week C57BL/6NCrl	fear conditioning by electric foot shock	in drinking water (20 mg/kg/day) 12 h after conditioning and for 28 days, followed by 28-day wash out	dorsal cornu ammonis 1 (CA1) 9 weeks after stress	N = 9: 3 NS-Veh; 3 S-Veh; 3 S-FLX	Illumina Genome Analyzer II system
	GSE43261	[51]	7–8-week C57BL/6J	corticosterone (35 μg/mL) in drinking water for 21 days	in drinking water (160 μg/mL) for 21 days	dorsal or ventral dentate gyrus	N = 30: (8 NS-Veh; 7 S-FLX-R) per area	Affymetrix Mouse Genome 430 2.0
rat S-FLX vs. S-Veh	GSE56028	[52]	2-month Wistar	6-week UCMS	ip administration (10 mg/kg/day in ultra-pure water), for the last 2 weeks	dentate gyrus	N = 9: 3 NS-Veh; 3 S-Veh; 3 S-FLX	Affymetrix Rat Gene 1.0 ST
	SRP131063	NA	6–7-week Sprague-Dawley	5-week UCMS	ip administration (7 mg/kg/day) the last week of UCMS.	hippocampus	N = 9: 3 NS-Sal; 3 S-Sal; 3 S-FLX	Illumina HiSeq 4000
	SRP084288	[53]	6-week Sprague-Dawley	restraint stress (6 h/day) for 28 days	administration by gavage (10 mg/kg/day) one hour before stress for 28 days	hippocampus	N = 3: 1 CTL; 1 S; 1 S-FLX	Illumina Nextseq 500/151 nt
mouse FLX vs. Veh	GSE84185	[21]	8-week BALB/c	No	In drinking water (10–20 mg/kg/day) for 6 weeks	dentate gyrus	N= 16: 8 NS-Veh; 8 NS-FLX	Agilent SurePrint G3 Mouse Gene Expression 8 × 60 K
	SRP057486	[48]	7–8-week C57BL/6NCrl	No	in drinking water (20 mg/kg/day) for 28 days, followed by 5-week wash out	dorsal cornu ammonis 1 (CA1)	N = 6: 3 Veh; 3 FLX	Illumina Genome Analyzer II system
	GSE118669	[54]	9-week C57BL/6J	No	subcutaneous releasing pellet (15 mg/kg/day) in the dorsal interscapular region for 3 weeks	dentate gyrus	N = 16: 8 Veh; 8 FLX	Affymetrix Mouse Genome 430 2.0
	GSE54307	[55]	8-week C57BL/6J	No	in drinking water (22 mg/kg/day) for 4 weeks	dentate gyrus 24 h after the end of FLX treatment	N = 2: 1 Veh; 1 FLX	Affymetrix Mouse Genome 430 2.0
	GSE6476	[56]	3–5-week DBA/2J	No	in drinking water (18 mg/kg/day) for 3 weeks	hippocampus	N = 4: 2 CTL; 2 FLX	Affymetrix Mouse Genome 430 2.0
rat FLX vs. Veh	SRP056480	[57]	Wistar	No	oral gavage (12 mg/kg/day) or Veh (1% methylcellulose) from PND 67 to 88	hippocampus at PND 128	N = 4: 2 pools of 2 rats per group (Veh; FLX)	Illumina Genome Analyzer IIx
	GSE42940	[58]	2-month Sprague-Dawley	No	po (10 mg/kg) or Veh (5% sucrose) from PND 2 to 21 and wash out for 5 weeks	hippocampus	N = 8: 4 Veh; 4 FLX	Agilent TIFR-Rat-8 × 15 K v1.0

CTL, control; FLX, fluoxetine; ip, intraperitoneal; NS, non-stressed; po, per os; S, stressed; Sal, saline; UCMS, unpredictable chronic mild stress, Veh, vehicle.

**Table 2 ijms-23-13543-t002:** Signature of FLX effect in stressed animals (integration method).

Dataset ID	GSE43261		GSE56028		GSE84185		SRP057486		SRP084288		SRP131063		Surget 2009		Consensus Score
Gene	log_2_FC	*p*	log_2_FC	*p*	log_2_FC	*p*	log_2_FC	*p*	log_2_FC	*p*	log_2_FC	*p*	log_2_FC	*p*	
*Adprm*	−0.096	**4.25 × 10^−2^**	−0.251	**5.43 × 10^−3^**	−0.105	**4.01 × 10^−2^**	0.035	6.92 × 10^−1^	−**0.287**	**7.68 × 10^−3^**	0.054	6.83 × 10^−1^	−0.012	9.11 × 10^−1^	−4
*Arc*	**1.796**	**9.70 × 10^−3^**	**−0.379**	**3.20 × 10^−2^**	**1.946**	**5.65 × 10^−6^**	−0.011	7.98 × 10^−1^	−0.001	9.61 × 10^−1^	**0.282**	**3.87 ×** **10^−4^**	**−0.390**	**2.18 ×** **10^−2^**	1
*Ddah1*	**0.370**	**1.03 × 10^−4^**	−0.051	6.22 × 10^−1^	0.180	**1.58 × 10^−2^**	0.104	2.28 × 10^−1^	0.192	**1.96 × 10^−2^**	−0.205	**1.75 ×** **10^−3^**	0.174	**2.51 ×** **10^−2^**	3
*Egr1*	**1.434**	**5.36 × 10^−3^**	−0.091	4.48 × 10^−1^	**1.883**	**1.13 × 10^−5^**	**−0.308**	**5.97 × 10^−4^**	−0.117	4.20 × 10^−1^	0.255	**1.48 ×** **10^−3^**	**−0.286**	**2.65 ×** **10^−2^**	1
*Ephb6*	**1.084**	**8.54 × 10^−12^**	−0.176	7.84 × 10^−2^	**1.328**	**3.22 × 10^−8^**	−0.034	7.02 × 10^−1^	0.242	**4.84 × 10^−3^**	**0.269**	**1.71 ×** **10^−4^**	−0.037	7.49 × 10^−1^	4
*Hmgcs1*	**0.766**	**7.36 × 10^−5^**	−0.218	**3.04 × 10^−2^**	**0.336**	**4.46 × 10^−2^**	−0.145	7.06 × 10^−2^	0.089	1.75 × 10^−1^	**−0.410**	**3.43 ×** **10^−11^**	−0.217	**1.73 ×** **10^−2^**	−1
*Kcng2*	**−0.750**	**2.29 × 10^−14^**	0.146	**4.95 × 10^−2^**	**−1.648**	**1.07 × 10^−5^**	0.094	2.29 × 10^−1^	0.238	**2.84 × 10^−2^**	0.127	**4.95 ×** **10^−^** ** ^2^ **	0.075	5.17 × 10^−1^	1
*Klhl5*	**−0.394**	**2.91 × 10^−4^**	−0.073	3.34 × 10^−1^	**−0.562**	**1.95 × 10^−5^**	−0.013	8.71 × 10^−1^	0.242	**4.04 × 10^−2^**	−0.172	**8.79 ×** **10^−3^**	−0.177	**2.34 ×** **10^−2^**	−3
*Lzts1*	**0.301**	**1.35 × 10^−3^**	0.014	8.95 × 10^−1^	**0.723**	**6.19 × 10^−4^**	−0.020	8.23 × 10^−1^	**0.345**	**7.62 × 10^−5^**	0.165	**3.34 ×** **10^−2^**	−0.096	5.85 × 10^−1^	4
*Mef2d*	0.202	**5.24 × 10^−4^**	0.038	7.29 × 10^−1^	**0.646**	**1.82 × 10^−7^**	0.042	6.25 × 10^−1^	0.227	**2.63 × 10^−2^**	0.263	**1.14 ×** **10^−^** ** ^5^ **	−0.101	1.49 × 10^−1^	4
*Nfib*	−0.224	**4.91 × 10^−3^**	0.139	6.01 × 10^−2^	**−0.929**	**4.53 × 10^−7^**	−0.052	5.40 × 10^−1^	**−0.329**	**4.12 × 10^−4^**	−0.062	4.02 × 10^−1^	**−0.310**	**1.20 ×** **10^−2^**	−4
*Nr4a1*	**1.390**	**2.01 × 10^−2^**	**−0.395**	**4.02 × 10^−2^**	**1.906**	**1.79 × 10^−4^**	−0.201	**2.30 × 10^−2^**	−0.072	9.00 × 10^−1^	**0.352**	**1.08 ×** **10^−^** ** ^5^ **	**−0.303**	**2.63 ×** **10^−2^**	0
*Oxtr*	**1.420**	**1.52 × 10^−5^**	0.214	**4.93 × 10^−2^**	**−1.121**	**5.44 × 10^−4^**	0.138	**4.23 × 10^−2^**	−0.012	8.13 × 10^−1^	0.091	**4.18 ×** **10^−2^**	−0.051	7.59 × 10^−1^	3
*Ppara*	−0.165	**8.29 × 10^−3^**	−0.187	**3.29 × 10^−2^**	−0.160	**3.02 × 10^−2^**	−0.024	7.45 × 10^−1^	−0.045	5.08 × 10^−1^	−0.101	4.22 × 10^−1^	−0.175	**2.86 ×** **10^−2^**	−4
*Prkar1b*	**0.850**	**6.71 × 10^−12^**	−0.018	8.51 × 10^−1^	**0.774**	**3.31 × 10^−7^**	−0.005	9.52 × 10^−1^	**0.284**	**3.64 × 10^−4^**	0.181	**9.95 ×** **10^−3^**	−0.148	1.14 × 10^−1^	4
*Rimbp2*	**0.622**	**1.40 × 10^−6^**	−0.121	2.24 × 10^−1^	**0.768**	**9.56 × 10^−3^**	0.179	**3.87 × 10^−2^**	0.048	4.99 × 10^−1^	0.158	**7.54 ×** **10^−3^**	−0.001	9.95 × 10^−1^	4
*Rps10*	−0.179	**7.62 × 10^−5^**	0.000	1.00	−0.196	**4.56 × 10^−3^**	−0.008	9.29 × 10^−1^	−0.227	**3.07 × 10^−3^**	−0.253	**3.39 ×** **10^−4^**	−0.194	8.19 × 10^−2^	−4
*Sgsm2*	0.230	**4.56 × 10^−4^**	−0.077	3.07 × 10^−1^	0.180	**2.16 × 10^−2^**	0.201	**2.30 × 10^−2^**	0.164	6.88 × 10^−2^	0.123	**4.94 ×** **10^−2^**	−0.048	4.99 × 10^−1^	4
*Ttbk1*	0.209	**2.86 ×** **10** ** ^−4^ **	−0.093	3.12 × 10^−1^	**0.386**	**1.58 ×** **10^−^** ** ^6^ **	−0.019	8.15 × 10^−1^	0.233	**9.14 × 10^−3^**	**0.272**	**1.71 ×** **10^−4^**	−0.151	1.97 × 10^−1^	4
*Ttyh3*	**0.275**	**1.01 ×** **10^−^** ** ^4^ **	−0.065	3.24 × 10^−1^	**0.400**	**1.68 ×** **10^−^** ** ^5^ **	0.035	6.68 × 10^−1^	0.249	**3.61 × 10^−3^**	0.231	**1.14 × 10^−3^**	0.061	4.37 × 10^−1^	4
*Wnk1*	−0158	**1.17 ×** **10^−^** ** ^3^ **	0159	2.04 × 10^−1^	**−0300**	**2.07 ×** **10^−^** ** ^2^ **	−0183	**4.11 ×** **10^−^** ** ^2^ **	−0223	**2.17 ×** **10^−^** ** ^3^ **	−0032	5.92 × 10^−1^	−0089	2.90 × 10^−1^	−4
*Wtap*	−0137	**3.81 ×** **10^−^** ** ^3^ **	0004	9.58 × 101	**−0271**	**4.47 ×** **10^−^** ** ^3^ **	0105	**2.29 ×** **10^−^** ** ^1^ **	−0026	**8.53 × 10^−1^**	−0126	**4.91 ×** **10^−^** ** ^2^ **	−0254	**2.83 ×** **10^−^** ** ^2^ **	−4
*Zfhx2*	**−0.350**	**2.05 × 10^−7^**	−0.165	1.51 × 10^−1^	**−0.421**	**4.88 × 10^−3^**	−0.207	**1.69 × 10^−2^**	−0.200	**1.95 × 10^−2^**	**0.347**	**7.94 × 10^−7^**	**−0.270**	**3.20 × 10^−3^**	−4

|log_2_FC| > 0.263 and *p*-values < 0.05 are in bold. Consensus score indicates whether FLX induces mainly upregulation (positive score) or downregulation (negative score) of gene expression.

**Table 3 ijms-23-13543-t003:** Signature of FLX effect in stressed animals (portrait score method).

Dataset ID	GSE43261		GSE56028		GSE84185		SRP057486		SRP084288		SRP131063		Surget 2009		Consensus Score	Portrait Score
Gene	log_2_FC	*p*	log_2_FC	*p*	log_2_FC	*p*	log_2_FC	*p*	log_2_FC	*p*	log_2_FC	*p*	log_2_FC	*p*		
*Ier5*	**1.143**	**1.69 × 10^−11^**	−0.009	9.29 × 10^−1^	**1.786**	**1.39 × 10^−5^**	0.107	2.33 × 10^−1^	0.119	3.36 × 10^−1^	0.157	**2.00 × 10^−2^**	0.238	8.61 × 10^−2^	3	4.5566666
*Mapk4*	**0.928**	**9.14 × 10^−7^**	−0.040	6.26 × 10^−1^	**1.021**	**3.20 × 10^−5^**	−0.015	8.55 × 10^−1^	0.169	7.03 × 10^−2^	0.176	**1.66 × 10^−3^**	0.150	2.30 × 10^−1^	3	4.5555544
*Diras2*	**0.310**	**6.90 × 10^−7^**	−0.046	5.63 × 10^−1^	**0.387**	**4.61 × 10^−7^**	0.110	2.15 × 10^−1^	−0.028	4.61 × 10^−1^	0.212	**8.37 × 10^−4^**	0.121	3.74 × 10^−1^	3	4.4544333
*Kcnh3*	0.162	**3.22 × 10^−2^**	−0.001	9.90 × 10^−1^	**1.004**	**2.57 × 10^−7^**	0.169	5.02 × 10^−2^	0.189	**3.49 × 10^−2^**	**0.284**	**1.83 × 10^−4^**	0.048	5.57 × 10^−1^	4	4.4456666
*Sema7a*	**0.786**	**7.29 × 10^−6^**	0.056	5.72 × 10^−1^	**1.083**	**1.89 × 10^−6^**	0.037	6.38 × 10^−1^	0.008	8.35 × 10^−1^	0.168	**1.43 × 10^−2^**	0.158	1.86 × 10^−1^	3	4.4444666
*Mapk9*	**0.350**	**1.45 × 10^−6^**	0.026	6.68 × 10^−1^	0.248	**1.13 × 10^−4^**	−0.028	7.22 × 10^−1^	0.218	**5.68 × 10^−3^**	−0.031	5.78 × 10^−1^	0.132	1.41 × 10^−1^	3	4.4444443
*Nptx2*	**2.410**	**2.74 × 10^−7^**	−0.132	5.64 × 10^−1^	**1.804**	**3.80 × 10^−5^**	−0.093	2.98 × 10^−1^	0.209	**3.74 × 10^−2^**	0.237	**8.39 × 10^−4^**	0.069	5.27 × 10^−1^	4	4.3334333
** *Prkar1b* **	**0.850**	**6.71 × 10^−12^**	−0.018	8.51 × 10^−1^	**0.774**	**3.31 × 10^−7^**	−0.005	9.52 × 10^−1^	**0.284**	**3.64 × 10^−4^**	0.181	**9.95 × 10^−3^**	−0.148	1.14 × 10^−1^	4	4.3333333
** *Hmgcs1* **	**0.766**	**7.36 × 10^−5^**	−0.218	**3.04 × 10^−2^**	**0.336**	**4.46 × 10^−2^**	−0.145	7.06 × 10^−2^	0.089	1.75 × 10^−1^	**−0.410**	**3.43 × 10^−11^**	−0.217	**1.73 × 10^−2^**	−1	−4.3211111
*Rspo3*	**−0.630**	**1.34 × 10^−6^**	−0.163	8.89 × 10^−2^	**−0.966**	**3.40 × 10^−6^**	0.063	4.81 × 10^−1^	**−0.269**	**2.10 × 10^−2^**	0.072	4.25 × 10^−1^	−0.099	4.70 × 10^−1^	−3	−4.4432333
*Cnn3*	0.155	**2.09x 10^−2^**	−0.029	7.31 × 10^−1^	**−0.772**	**1.08 × 10^−6^**	−0.055	5.13 × 10^−1^	−0.228	**9.77 × 10^−3^**	−0.175	**3.09 × 10^−3^**	−0.183	8.63 × 10^−2^	−2	−4.4434445
*Sh3d19*	**−0.677**	**1.80 × 10^−11^**	−0.020	8.19 × 10^−1^	**−0.643**	**2.80 × 10^−6^**	−0.112	1.86 × 10^−1^	−0.010	9.62 × 10^−1^	−0.131	1.16 × 10^−1^	**−0.313**	5.57 × 10^−2^	−2	−4.4555555

|log_2_FC| > 0.263 and *p*-values < 0.05 are in bold. Genes in bold are also in Table 2. Consensus score indicates whether FLX induces mainly upregulation (positive score) or downregulation (negative score) of gene expression.

**Table 4 ijms-23-13543-t004:** Ontological analysis of upregulated genes (consensus and portrait scores ≥ 2) after FLX treatment in stressed animals.

Category	ID	Name	Genes	Count	Fold enrichment	*Padj*
Cellular component	GO:0045202	synapse	*Add1*, *Adgrb1*, *Adgrl1*, *Arhgap44*, *Arhgdia*, *Atcay*, *Baalc*, *Btbd8*, *Cacna1g*, *Cacnb3*, *Cacng2*, *Cdkl5*, *Clstn1*, *Cyp46a1*, *Dlg1*, *Dlg2*, *Dlgap3*, *Dnajc6*, *Drd1*, *Egr3*, *Gabbr1*, *Gabra1*, *Gabrg2*, *Git1*, *Grin1*, *Grk2*, *Iqsec3*, *Kcnb1*, *Kcnk1*, *Lrrc4b*, *Magi2*, *Mink1*, *Mthfr*, *Ncs1*, *Nlgn2*, *Ntrk2*, *Pak5*, *Palm*, *Ppm1h*, ***Prkar1b***, *Psd3*, *Rims3*, *Sh2d5*, *Shank1*, *Shank3*, *Slc30a3*, *Slc4a10*, *Slc5a7*, *Slc6a17*, *Strn4*, *Stx1a*, *Sv2a*, *Syn1*, *Syndig1*, *Syngap1*, *Syngr1*, *Vgf*, *Znrf1*	58	3.61	2.05 × 10^−14^
	GO:0014069	postsynaptic density	*Add1*, *Adgrb1*, *Adgrl1*, *Arhgap44*, *Arhgef2*, *Atp1a1*, *Baalc*, *Baiap2*, *Cacng2*, *Cdk5r1*, *Clstn1*, *Dclk1*, *Dlg1*, *Dlg2*, *Dlgap3*, *Dnajc6*, *Git1*, *Grin1*, *Iqsec3*, *Lrp8*, *Magi2*, *Mink1*, *Ncs1*, *Ntrk2*, *Palm*, *Psd3*, *Sh2d5*, *Shank1*, *Shank3*, *Shisa8*, *Stx1a*, *Syn1*, *Syndig1*, *Syngap1*	34	5.15	1.31 × 10^−11^
	GO:0030054	cell junction	*Adgrb1*, *Adgrl1*, *Arhgap44*, *Arhgef2*, *Atcay*, *Baalc*, *Basp1*, *Btbd8*, *Cacng2*, *Clstn1*, *Cyp46a1*, *Dlg1*, *Dlg2*, *Dlgap3*, *Drd1*, *Egr3*, *Gabbr1*, *Gabra1*, *Gabrg2*, *Git1*, *Grin1*, *Iqsec3*, *Itgav*, *Kcnb1*, *Kcnk1*, *Lrrc4b*, *Magi2*, *Mink1*, *Mpp7*, *Ncs1*, *Nlgn2*, *Ntrk2*, *Palm*, *Psd3*, *Ptk2*, ***Rimbp2***, *Rims3*, *Sh2d5*, *Shank1*, *Shank3*, *Slc30a3*, *Slc4a10*, *Slc5a7*, *Slc6a17*, *Sptbn2*, *Strn4*, *Stx1a*, *Sv2a*, *Syndig1*, *Syngap1*, *Syngr1*, *Syt5*, *Vasp*, *Znrf1*	53	3.13	2.12 × 10^−10^
	GO:0005891	voltage-gated calcium channel complex	*Cacna1g*, *Cacna2d2*, *Cacna2d4*, *Cacnb1*, *Cacnb3*, *Cacng2*,	6	13.1	3.40 × 10^−2^
Biological process	GO:0035556	positive regulation of synaptic transmission, glutamatergic	*Cacng2*, *Cacng3*, *Drd1*, *Grin1*, *Iqsec2*, *Nlgn2*, *Ntrk2*, ***Oxtr***, *Ptgs2*, *Shank3*, *Tnr*	11	15.6	2.72 × 10^−6^
	GO:0010807	regulation of synaptic vesicle priming	*Nabp*, *Stx1a*, *Stx1b*, *Stxbp1*, *Stxbp5*	5	29.9	2.90 × 10^−2^
KEGG pathway	mmu04921	oxytocin signaling pathway	*Adcy2*, *Cacna2d2*, *Cacna2d4*, *Cacnb1*, *Cacnb3*, *Cacng2*, *Cacng3*, *Camk1g*, *Camk2d*, *Kras*, *Mylk3*, ***Oxtr***, *Pik3r6*, *Prkacb*, *Ptgs2*, *Rock2*	16	5.76	2.30 × 10^−5^
	mmu05414	dilated cardiomyopathy	*Adcy2*, *Cacna2d2*, *Cacna2d4*, *Cacnb1*, *Cacnb3*, *Cacng2*, *Cacng3*, *Itgav*, *Lmna*, *Prkacb*, *Tgfb3*	11	6.44	1.66 × 10^−3^
	mmu04261	adrenergic signaling in cardiomyocytes	*Adcy2*, *Atp1a1*, *Cacna2d2*, *Cacna2d4*, *Cacnb1*, *Cacnb3*, *Cacng2*, *Cacng3*, *Camk2d*, *Mapk11*, *Pik3r6*, *Ppp2r2c*, *Prkacb*	13	4.71	4.43 × 10^−3^
	mmu04722	neurotrophin signaling pathway	*Arhgdia*, *Arhgdig*, *Bdnf*, *Camk2d*, *Grb2*, *Kras*, ***Mapk9***, *Mapk11*, *Ntrk2*, *Pik3r2*, *Sort1*	11	5.01	1.52 × 10^−2^
Molecular function	GO:0005245	voltage-gated calcium channel activity	*Cacna1g*, *Cacna2d2*, *Cacna2d4*, *Cacnb1*, *Cacnb3*, *Cacng2*, *Cacng3*, *Itgav*, *Ncs1*	9	12.0	3.93 × 10^−4^
	GO:0016301	kinase activity	*Acvr1c*, *Brsk2*, *Camk1g*, *Camk2d*, *Cdk5r1*, *Cdkl5*, *Dclk1*, *Grk2*, *Hkdc1*, *Ikbkg*, *Itpkc*, *Lmtk2*, *Mapk11*, ***Mapk4***, ***Mapk9***, *Mark4*, *Melk*, *Mink1*, *Mylk3*, *Ntrk2*, *Pak5*, *Pfkm*, *Pim1*, *Pip4k2b*, *Prkacb*, ***Prkar1b***, *Ptk2*, *Rock2*, *Tesk1*, *Ttbk1*, *Tyro3*	31	2.49	4.97 × 10^−3^
	GO:0004672	protein kinase activity	*Acvr1c*, *Brsk2*, *Camk1g*, *Camk2d*, *Cdkl5*, *Dclk1*, ***Ephb6***, *Grk2*, *Lmtk2*, *Mapk11*, ***Mapk4***, ***Mapk9***, *Mark4*, *Melk*, *Mink1*, *Mylk3*, *Ntrk2*, *Pak5*, *Pim1*, *Prkacb*, *Ptk2*, *Rock2*, *Tesk1*, *Ttbk1*, *Tyro3*	25	2.56	3.01 × 10^−2^

Genes belonging to the core signatures (Table 2 and Table 3) are in bold.

**Table 5 ijms-23-13543-t005:** Ontological analysis (DAVID) of downregulated genes (consensus and portrait scores < −2) after FLX treatment in stressed animals.

Category	ID	Name	Genes	Count	Fold Enrichment	*Padj*
Cellular component	GO:0022626	cytosolic ribosome	*Apod*, *Eif2ak4*, *Rpl4*, *Rpl11*, *Rpl17*, *Rpl21*, *Rpl24*, *Rpl26*, *Rpl31*, *Rpl34*, *Rps6*, *Rps7*, ***Rps10***, *Rps11*, *Rps12*, *Rps13*, *Rps15a*, *Rps24*, *Rps27a*, *Uba52*	20	13.9	4.75 × 10^−14^
	GO:0005840	ribosome	*Mrpl3*, *Mrpl50*, *Mrps35*, *Rpl4*, *Rpl11*, *Rpl17*, *Rpl21*, *Rpl24*, *Rpl31*, *Rpl34*, *Rps6*, *Rps7*, ***Rps10***, *Rps11*, *Rps12*, *Rps13*, *Rps15a*, *Rps24*, *Rps27a*, *Uba52*	20	5.76	9.32 × 10^−7^
	GO:0005912	adherens junction	*Ahi1*, *Cdh2*, *Dlg5*, *Ctnna1*, *Dll1*, *Fermt2*, *Frmd4a*, *Frmd4b*, *Pdlim5*, *Pgm5*, *Pkp2*, *Pkp5*, *Tspan33*	13	4.61	1.14 × 10^−2^
Molecular function	G:0003735	structural constituent of ribosome	*Mrpl3*, *Mrps35*, *Rpl4*, *Rpl11*, *Rpl17*, *Rpl21*, *Rpl24*, *Rpl26*, *Rpl31*, *Rpl34*, *Rps6*, *Rps7*, ***Rps10***, *Rps11*, *Rps12*, *Rps13*, *Rps15a*, *Rps24*, *Rps27a*, *Uba52*	20	5.85	9.53 × 10^−7^
Biological process	GO:0006412	translation	*Eif2b2*, *Eif2s2*, *Eif3m*, *Mrpl3*, *Rpl4*, *Rpl11*, *Rpl17*, *Rpl21*, *Rpl24*, *Rpl26*, *Rpl31*, *Rpl34*, *Rps6*, *Rps7*, *Rps11*, *Rps12*, *Rps13*, *Rps15a*, *Rps24*, *Rps27a*, *Tars2*, *Uba52*	22	3.91	5.58 × 10^−4^

Genes belonging to the core signatures (Table 2 and Table 3) are in bold.

**Table 6 ijms-23-13543-t006:** Signature of FLX effect in naïve animals (integration method).

Dataset ID	GSE118669		GSE42940		GSE4307		GSE6476		GSE84185		SRP056481		SRP057486		Consensus Score
Gene	log_2_FC	*p*	log_2_FC	*p*	log_2_FC	*p*	log_2_FC	*p*	log_2_FC	*p*	log_2_FC	*p*	log_2_FC	*p*	
*Arhgef28*	**−0.354**	**1.14 × 10^−3^**	**−0.774**	**1.60 × 10^−2^**	**−0.748**	**8.14 × 10^−2^**	−0.191	**3.42 × 10^−2^**	**−1.251**	**7.89 × 10^−11^**	0.140	2.15 × 10^−1^	−0.122	2.92 × 10^−1^	−4
*Arrb2*	−0.179	**1.94 × 10^−2^**	**−1.311**	**1.48 × 10^−3^**	−0.184	5.83 × 10^−1^	−0.156	5.25 × 10^−2^	−0.250	**2.34 × 10^−3^**	−0.013	9.13 × 10^−1^	−0.227	**4.81 × 10^−2^**	−4
*Cd68*	**0.377**	**3.33 × 10^−4^**	−0.257	1.77 × 10^−1^	**0.9468**	1.94 × 10^−1^	**0.842**	**3.67 × 10^−4^**	0.193	**4.49 × 10^−2^**	0.0034	9.49 × 10^−1^	**0.352**	**1.03 × 10^−2^**	4
*Cfh*	0.077	6.62 × 10^−1^	**0.461**	**2.91 × 10^−2^**	**1.3316**	**2.28 × 10^−2^**	**0.430**	**8.38 × 10^−4^**	0.027	3.68 × 10^−1^	0.054	5.75 × 10^−1^	**0.392**	**1.96 × 10^−3^**	4
*Cdon*	**−0.274**	**1.76 × 10^−2^**	**−0.847**	**4.12 × 10^−2^**	**−3.139**	**3.14 × 10^−3^**	**−0.359**	7.20 × 10^−2^	**−0.607**	**1.71 × 10^−4^**	−0.103	3.78 × 10^−1^	−0.031	8.19 × 10^−1^	−4
*Chgb*	0.227	**3.03 × 10^−2^**	**−0.751**	**4.05 × 10^−2^**	**0.507**	3.48 × 10^−1^	**0.396**	**1.27 × 10^−3^**	**−0.376**	**1.84 × 10^−5^**	−0.249	**3.94 × 10^−2^**	0.111	3.34 × 10^−1^	−1
*Ddr1*	0.245	**9.08 × 10^−4^**	**−0.276**	2.83 × 10^−1^	**1.145**	**5.00 × 10^−2^**	0.261	**8.30 × 10^−4^**	**−0.493**	**2.22 × 10^−5^**	0.097	4.09 × 10^−1^	**0.342**	**2.84 × 10^−3^**	3
*Doc2b*	**−0.506**	**1.51 × 10^−3^**	**0.371**	1.83 × 10^−1^	**−0.798**	7.63 × 10^−2^	−0.233	**1.07 × 10^−2^**	**−3.232**	**3.42 × 10^−11^**	0.028	7.90 × 10^−1^	**−0.416**	**4.12 × 10^−4^**	−4
*Fat4*	**−0.890**	**4.54 × 10^−4^**	0.000	1.00	0.000	1.00	**−1.352**	**1.80 × 10^−3^**	**−1.748**	**2.48 × 10^−11^**	−0.210	6.03 × 10^−2^	**−0.342**	**1.25 × 10^−2^**	−4
*Gpr12*	**−0.402**	**3.06 × 10^−2^**	**−0.708**	**2.38 × 10^−2^**	**−1.981**	1.71 × 10^−1^	**−0.837**	**2.38 × 10^−4^**	**−0.491**	**1.25 × 10^−6^**	−0.115	3.16 × 10^−1^	−0.001	9.95 × 10^−1^	−4
*Gsn*	**0.303**	**2.82 × 10^−2^**	**0.783**	1.49 × 10^−1^	**1.488**	7.55 × 10^−2^	**0.549**	**4.24 × 10^−3^**	**0.397**	**4.04 × 10^−6^**	0.106	3.13 × 10^−1^	**0.495**	**1.16 × 10^−4^**	4
*H2-D1*	0.210	9.67 × 10^−2^	0.000	1.00	**1.173**	**3.25 × 10^−2^**	**0.673**	**1.07 × 10^−2^**	**0.424**	**3.22 × 10^−4^**	0.000	1.00	**0.425**	**3.57 × 10^−4^**	4
*Homer1*	0.174	**4.03 × 10^−2^**	**0.688**	**5.84 × 10^−3^**	**0.873**	6.39 × 10^−2^	**2.919**	**1.41 × 10^−3^**	**1.133**	**7.05 × 10^−12^**	0.057	6.17 × 10^−1^	**−0.307**	**2.55 × 10^−2^**	3
*Htr1b*	0.121	**3.19 × 10^−2^**	**0.544**	**9.07 × 10^−3^**	**1.466**	**1.88 × 10^−2^**	**−1.189**	**1.03 × 10^−3^**	**1.084**	**2.35 × 10^−3^**	0.076	3.90 × 10^−1^	−0.262	5.54 × 10^−2^	3
*Htr5b*	**0.920**	**3.29 × 10^−3^**	**0.987**	7.01 × 10^−2^	**0.847**	7.59 × 10^−1^	**−1.112**	**3.30 × 10^−2^**	**1.252**	**7.36 × 10^−3^**	0.224	**4.37 × 10^−2^**	**−0.477**	**4.97 × 10^−4^**	1
*Igfbp6*	**1.110**	**2.04 × 10^−4^**	0.081	6.67 × 10^−1^	**2.796**	2.01 × 10^−1^	**0.821**	**1.64 × 10^−2^**	**2.720**	**3.41 × 10^−10^**	−0.079	4.68 × 10^−1^	**0.325**	**6.21 × 10^−3^**	4
*Ints10*	−0.129	**2.54 × 10^−2^**	0.000	1.00	0.000	1.00	**−0.562**	**6.79 × 10^−4^**	−0.259	**2.41 × 10^−3^**	0.016	8.92 × 10^−1^	−0.254	**1.42 × 10^−2^**	−4
** *Isoc1* **	**−0.460**	**6.82 × 10^−5^**	**−0.337**	**4.55 × 10^−2^**	**−1.803**	**1.54 × 10^−2^**	**−0.360**	**2.98 × 10^−2^**	**−0.761**	**1.26 × 10^−7^**	−0.051	4.85 × 10^−1^	0.035	7.98 × 10^−1^	**−5**
*Itga4*	**−0.437**	**1.33 × 10^−2^**	−0.007	9.83 × 10^−1^	**−3.476**	6.77 × 10^−2^	**−0.601**	**2.59 × 10^−2^**	**−1.608**	**1.14 × 10^−10^**	−0.012	8.44 × 10^−1^	**−0.490**	**2.28 × 10^−5^**	−4
*Itsn1*	−0.238	**3.80 × 10^−3^**	**−2.471**	**1.91 × 10^−3^**	**−0.915**	5.91 × 10^−2^	−0.232	**2.93 × 10^−3^**	**−0.785**	**1.52 × 10^−8^**	−0.068	5.71 × 10^−1^	−0.164	1.84 × 10^−1^	−4
*Kirrel3*	**−0.264**	**1.02 × 10^−3^**	**−0.603**	**4.93 × 10^−2^**	0.000	1.00	**−0.447**	**5.77 × 10^−3^**	**−0.617**	**9.80 × 10^−8^**	−0.105	3.65 × 10^−1^	0.030	8.29 × 10^−1^	−4
*Knstrn*	0.182	**2.56 × 10^−2^**	0.000	1.00	**3.322**	**2.93 × 10^−3^**	0.153	**1.00 × 10^−2^**	**1.627**	**1.10 × 10^−11^**	0.002	9.38 × 10^−1^	0.026	8.34 × 10^−1^	4
** *Map1a* **	−0.146	**1.24 × 10^−2^**	**−3.255**	**4.42 × 10^−3^**	0.000	1.00	−0.116	**2.02 × 10^−2^**	−0.199	**2.12 × 10^−3^**	−0.059	6.26 × 10^−1^	**−0.273**	**1.40 × 10^−2^**	**−5**
*Mpdz*	**−0.355**	**9.19 × 10^−5^**	−0.074	7.18 × 10^−1^	−0.226	5.95 × 10^−1^	**−0.323**	**2.64 × 10^−2^**	**−0.540**	**6.63 × 10^−5^**	−0.071	5.55 × 10^−1^	−0.250	**4.02 × 10^−2^**	−4
*Mat2a*	0.195	**6.33 × 10^−3^**	**0.689**	**3.65 × 10^−3^**	**1.323**	**2.40 × 10^−2^**	−0.210	**1.34 × 10^−2^**	**−0.398**	**8.29 × 10^−5^**	0.046	6.98 × 10^−1^	0.014	8.81 × 10^−1^	1
*Mylk*	**0.333**	**3.40 × 10^−2^**	**−0.616**	**2.80 × 10^−3^**	**1.305**	2.24 × 10-1	**0.792**	**9.15 × 10^−3^**	**−1.215**	**1.07 × 10^−7^**	0.034	7.68 × 10^−1^	**0.455**	**8.35 × 10^−4^**	1
** *Myo1e* **	0.203	**9.58 × 10^−3^**	**0.540**	1.79 × 10^−1^	**1.855**	**1.08 × 10-2**	0.054	**3.40 × 10^−2^**	**0.837**	**2.74 × 10^−6^**	−0.005	9.59 × 10^−1^	**0.290**	**2.78 × 10^−2^**	**5**
*Negr1*	−0.255	**9.34 × 10^−3^**	**0.472**	5.96 × 10^−1^	0.000	1.00	**−0.326**	**3.76 × 10^−3^**	**−0.588**	**3.19 × 10^−3^**	−0.246	**3.52 × 10^−2^**	0.023	8.60 × 10^−1^	−4
*Nhsl2*	**−0.427**	**5.59 × 10^−4^**	0.000	1.00	**−0.671**	3.28 × 10^−1^	**−0.356**	**1.84 × 10^−2^**	**−0.868**	**5.61 × 10^−10^**	0.000	1.00	**−0.302**	**6.20 × 10^−3^**	−4
*Ntrk3*	−0.253	**3.34 × 10^−3^**	0.179	2.81 × 10^−1^	**1.932**	5.89 × 10^−2^	**−0.156**	**5.25 × 10^−3^**	**−0.414**	**2.52 × 10^−6^**	−0.059	6.23 × 10^−1^	**−0.409**	**1.31 × 10^−4^**	−4
*Pcdh7*	**0.355**	**2.30 × 10^−4^**	**0.498**	3.95 × 10^−1^	**1.132**	**4.86 × 10^−2^**	0.222	**4.79 × 10^−2^**	**1.167**	**1.04 × 10^−7^**	−0.114	3.09 × 10^−1^	0.204	9.10 × 10^−2^	4
*Pcdh19*	**−0.572**	**4.44 × 10^−3^**	0.000	1.00	0.000	1.00	−0.252	**1.31 × 10^−2^**	**−1.419**	**9.69 × 10^−15^**	−0.114	3.35 × 10^−1^	**−0.413**	**1.08 × 10^−4^**	−4
*Pde4b*	**0.427**	**8.61 × 10^−6^**	−0.064	7.55 × 10^−1^	**0.763**	3.36 × 10^−1^	0.180	**2.91 × 10^−2^**	**0.485**	**7.66 × 10^−7^**	0.036	7.31 × 10^−1^	**0.517**	**6.00 × 10^−7^**	4
*Pde7b*	**−0.720**	**6.86 × 10^−4^**	0.063	8.82 × 10^−1^	**−1.251**	**3.96 × 10^−2^**	**−0.587**	**1.96 × 10^−2^**	**0.511**	**1.55 × 10^−3^**	**−0.287**	**8.38 × 10^−3^**	−0.054	6.10 × 10^−1^	−3
*Pdlim5*	−0.124	**2.95 × 10^−2^**	**−0.722**	**8.51 × 10^−3^**	**0.900**	7.04 × 10^−2^	**0.459**	**9.00 × 10^−3^**	**−2.857**	**2.37 × 10^−12^**	−0.165	9.09 × 10^−2^	**0.267**	**3.40 × 10^−2^**	−1
*Rab27a*	**−0.474**	**1.50 × 10^−3^**	**0.967**	1.82 × 10^−1^	**−2.928**	**5.78 × 10^−3^**	**−0.680**	**2.90 × 10^−2^**	**−1.773**	**8.63 × 10^−10^**	−0.047	6.45 × 10^−1^	−0.096	4.85 × 10^−1^	−4
*Rasgrf1*	**−0.394**	**5.36 × 10^−5^**	−0.200	2.02 × 10^−1^	**−0.443**	3.55 × 10^−1^	**−0.519**	**4.75 × 10^−4^**	**−1.051**	**2.38 × 10^−10^**	−0.006	9.59 × 10^−1^	**−0.335**	**5.16 × 10^−4^**	−4
*Rassf5*	**0.281**	**8.67 × 10^−3^**	**−1.470**	2.91 × 10^−1^	**1.934**	**9.78 × 10^−3^**	0.156	**8.56 × 10^−3^**	**0.398**	**6.61 × 10^−4^**	−0.004	9.66 × 10^−1^	0.225	8.59 × 10^−2^	4
*S100a6*	**0.473**	**4.52 × 10^−3^**	**−0.668**	**1.93 × 10^−2^**	**2.020**	1.08 × 10^−1^	**2.247**	**3.72 × 10^−4^**	**1.012**	**4.01 × 10^−11^**	−0.027	7.80 × 10^−1^	**0.368**	**7.22 × 10^−3^**	3
** *Scn3b* **	−0.245	**7.02 × 10^−3^**	**−0.479**	**1.05 × 10^−2^**	**−0.678**	1.15 × 10^−1^	**−0.594**	**5.20 × 10^−5^**	**−0.971**	**4.31 × 10^−8^**	0.178	1.41 × 10^−1^	**−0.304**	**2.13 × 10^−2^**	**−5**
*Sel1l3*	**0.281**	**8.20 × 10^−4^**	**−1.470**	1.00	**1.934**	**3.86 × 10^−2^**	0.156	**1.18 × 10^−2^**	**0.398**	**9.19 × 10^−7^**	−0.004	2.82 × 10^−1^	0.225	8.62 × 10^−1^	4
*Sema3a*	**0.948**	**6.47 × 10^−4^**	**−0.388**	4.91 × 10^−1^	**2.727**	**3.16 × 10^−2^**	**1.152**	**4.04 × 10^−3^**	**4.176**	**2.62 × 10^−11^**	−0.004	9.07 × 10^−1^	−0.084	5.41 × 10^−1^	4
*Sorcs1*	**1.183**	**1.18 × 10^−3^**	**−0.366**	3.72 × 10^−1^	**1.372**	9.98 × 10^−2^	**1.300**	**1.04 × 10^−2^**	**2.102**	**8.85 × 10^−8^**	0.059	5.70 × 10^−1^	**0.495**	**2.66 × 10^−4^**	4
*Tfrc*	**0.269**	**6.62 × 10^−4^**	**0.610**	3.00 × 10^−1^	**0.795**	2.65 × 10^−1^	**0.725**	**7.75 × 10^−3^**	**0.435**	**2.85 × 10^−6^**	−0.103	3.92 × 10^−1^	**0.298**	**8.19 × 10^−3^**	4
*Tnxb*	**−0.323**	**4.79 × 10^−4^**	0.000	1.00	**−0.695**	3.40 × 10^−1^	**−0.574**	**7.37 × 10^−3^**	**−1.462**	**6.94 × 10^−9^**	0.000	1.00	**−0.398**	**3.57 × 10^−3^**	−4
*Zfhx2*	−0.145	**1.02 × 10^−2^**	0.000	1.00	0.000	1.00	−0.051	**3.74 × 10^−2^**	**−0.653**	**2.51 × 10^−9^**	0.035	7.70 × 10^−1^	**−0.479**	**9.56 × 10^−5^**	−4
** *Zfp316* **	−0.218	**4.28 × 10^−3^**	**−1.324**	**3.00 × 10^−4^**	**−1.531**	**1.80 × 10^−2^**	**−0.912**	**8.63 × 10^−3^**	**−0.448**	**1.30 × 10^−3^**	−0.107	3.35 × 10^−1^	0.019	8.72 × 10^−1^	**−5**

|log_2_FC| > 0.263 and *p*-values ≤ 0.05 are in bold. Consensus score indicates whether FLX induces mainly upregulation (positive score) or downregulation (negative score) of gene expression. Genes with absolute value of consensus score equal to 5 are in bold.

**Table 7 ijms-23-13543-t007:** Signature of FLX effect in naïve animals (portrait score method).

Dataset ID	GSE118669		GSE42940		GSE4307		GSE6476		GSE84185		SRP056481		SRP057486		Consensus Score	Portrait Score
Gene	log_2_FC	*p*	log_2_FC	*p*	log_2_FC	*p*	log_2_FC	*p*	log_2_FC	*p*	log_2_FC	*p*	log_2_FC	*p*		
*Vip*	**0.306**	**3.19 × 10^−2^**	0.141	5.63 × 10^−1^	**0.934**	9.13 × 10^−2^	0.002	9.84 × 10^−1^	**2.129**	**6.92 × 10^−9^**	0.122	2.30 × 10^−1^	**0.318**	**1.61 × 10^−2^**	3	55,566,665
** *Gsn* **	**0.303**	**2.82 × 10^−2^**	**0.783**	1.49 × 10^−1^	**1.49**	7.55 × 10^−2^	**0.549**	**4.24 × 10^−3^**	**0.397**	**4.04 × 10^−6^**	0.106	3.13 × 10^−1^	**0.495**	**1.16 × 10^−4^**	4	46,777,777
*Cd9*	0.209	8.41 × 10^−2^	**0.458**	**2.96 × 10^−2^**	**1.346**	1.53 × 10^−1^	**0.764**	**3.18 × 10^−3^**	−0.168	5.47 × 10^−2^	0.205	8.59 × 10^−2^	**0.426**	**1.46 × 10^−3^**	3	46666655
*S100a10*	0.203	1.55 × 10^−1^	**0.308**	8.39 × 10^−2^	**1.600**	5.32 × 10^−2^	**0.619**	**1.09 × 10^−2^**	1.107	**9.75 × 10^−9^**	0.068	5.22 × 10^−1^	0.258	5.77 × 10^−2^	2	45,777,777
*Fcgr2b*	0.142	5.86 × 10^−2^	**0.688**	**3.57 × 10^−2^**	**1.566**	**3.08 × 10^−2^**	**0.920**	**1.59 × 10^−2^**	**0.617**	**7.66 × 10^−6^**	0.004	9.53 × 10^−1^	**0.390**	**3.44 × 10^−3^**	5	45,666,666
*Vsnl1*	0.212	**4.23 × 10^−2^**	**0.596**	**3.87 × 10^−2^**	**0.408**	2.37 × 10^−1^	−0.084	4.32 × 10^−1^	**1.372**	**4.34 × 10^−12^**	0.220	6.71 × 10^−2^	0.205	1.23 × 10^−1^	3	45,666,555
*Bgn*	0.144	4.83 × 10^−1^	**0.495**	**4.41 × 10^−2^**	**1.739**	6.86 × 10^−2^	**1.368**	**9.10 × 10^−3^**	**0.377**	3.47 × 10^−2^	0.100	3.76 × 10^−1^	**0.302**	**1.76 × 10^−2^**	4	45,555,777
** *Sel1l3* **	**0.281**	**8.20 × 10^−4^**	**−1.470**	1.00	**1.934**	**3.86 × 10^−2^**	0.156	**1.18 × 10^−2^**	**0.398**	**9.19 × 10^−7^**	−0.004	2.82 × 10^−1^	0.225	8.62 × 10^−1^	4	45,555,555
*Ppp2r5c*	**0.392**	**8.26 × 10^−4^**	**−0.289**	2.68 × 10^−1^	**1.23**	5.69 × 10^−2^	0.216	**7.36 × 10^−3^**	**1.25**	**3.61 × 10^−10^**	0.109	3.63 × 10^−1^	0.040	7.18 × 10^−1^	3	45,444,444
** *Ddr1* **	0.245	**9.08 × 10^−4^**	**−0.276**	2.83 × 10^−1^	**1.145**	**5.00 × 10^−2^**	0.261	**8.30 × 10^−4^**	**−0.493**	**2.22 × 10^−5^**	0.097	4.09 × 10^−1^	**0.342**	**2.84 × 10^−3^**	3	45,333,333
*Anxa5*	0.109	1.86 × 10^−1^	**0.678**	5.12 × 10^−2^	**0.981**	7.99 × 10^−2^	**0.641**	**2.84 × 10^−3^**	0.259	**3.55 × 10^−3^**	0.083	4.81 × 10^−1^	**0.314**	**9.22 × 10^−3^**	3	44,667,777
*Spock3*	**0.375**	**1.77 × 10^−2^**	**0.287**	3.05 × 10^−1^	**1.462**	**2.78 × 10^−2^**	**0.419**	**3.12 × 10^−3^**	**0.362**	**1.12 × 10^−2^**	0.053	6.58 × 10^−1^	**0.317**	**1.33 × 10^−2^**	5	44,557,777
*Icam1*	0.027	7.66 × 10^−1^	**0.640**	3.06 × 10^−1^	**0.869**	5.82 × 10^−2^	0.164	**6.84 × 10^−3^**	−0.008	5.96 × 10^−1^	0.110	2.43 × 10^−1^	**0.311**	**1.19 × 10^−2^**	2	44,555,556
*Rassf8*	**0.382**	**1.38 × 10^−2^**	**−0.501**	6.59 × 10^−1^	**1.424**	7.20 × 10^−2^	**0.577**	**2.20 × 10^−3^**	**0.971**	**1.39 × 10^−5^**	0.026	8.17 × 10^−1^	**0.279**	**3.82 × 10^−2^**	4	44,554,466
*Tspan5*	0.231	1.90 × 10^−1^	**1.609**	**1.56 × 10^−2^**	**1.041**	8.04 × 10^−2^	**0.378**	**3.79 × 10^−3^**	**0.787**	**2.00 × 10^−8^**	0.008	9.47 × 10^−1^	−0.108	3.79 × 10^−1^	3	44,554,444
*Kif5b*	0.218	**3.85 × 10^−3^**	**0.699**	**4.29 × 10^−2^**	**0.350**	2.86 × 10^−1^	0.055	4.41 × 10^−1^	**0.586**	**1.36 × 10^−8^**	0.012	9.17 × 10^−1^	0.257	**1.03 × 10^−2^**	4	44,455,557
* **Cfh** *	0.077	6.62 × 10^−1^	**0.461**	**2.91 × 10^−2^**	**1.3316**	**2.28 × 10^−2^**	**0.430**	**8.38 × 10^−4^**	0.027	3.68 × 10^−1^	0.054	5.75 × 10^−1^	**0.392**	**1.96 × 10^−3^**	4	44,455,556
** *Homer1* **	0.174	**4.03 × 10^−2^**	**0.688**	**5.84 × 10^−3^**	**0.873**	6.39 × 10^−2^	**2.919**	**1.41 × 10^−3^**	**1.133**	**7.05 × 10^−12^**	0.057	6.17 × 10^−1^	**−0.307**	**2.55 × 10^−2^**	3	44,455,555
*Tmem47*	**0.346**	**2.20 × 10^−3^**	0.000	1.00	**1.247**	**4.33 × 10^−2^**	**0.546**	**1.13 × 10^−2^**	0.000	1.00	0.050	6.67 × 10^−1^	0.250	**2.10 × 10^−2^**	4	44,445,555
*Tmem98*	0.190	**2.81 × 10^−2^**	0.000	1.00	**0.753**	7.81 × 10^−2^	**0.312**	2.47 × 10^−1^	−0.222	**2.87 × 10^−2^**	0.184	6.86 × 10^−2^	**0.655**	**1.19 × 10^−6^**	1	44,445,444
** *Knstrn* **	0.182	**2.56 × 10^−2^**	0.000	1.00	**3.322**	**2.93 × 10^−3^**	0.153	**1.00 × 10^−2^**	**1.627**	**1.10 × 10^−11^**	0.002	9.38 × 10^−1^	0.026	8.34 × 10^−1^	4	44,444,444
** *Sema3a* **	**0.948**	**6.47 × 10^−4^**	**−0.388**	4.91 × 10^−1^	**2.727**	**3.16 × 10^−2^**	**1.152**	**4.04 × 10^−3^**	**4.176**	**2.62 × 10^−11^**	−0.004	9.07 × 10^−1^	−0.084	5.41 × 10^−1^	4	44,443,312
*C1qb*	0.240	**5.04 × 10^−4^**	−0.180	3.60 × 10^−1^	**1.00**	**4.37 × 10^−2^**	**0.873**	**1.59 × 10^−4^**	−0.011	8.42 × 10^−1^	0.030	7.94 × 10^−1^	**0.473**	**2.69 × 10^−5^**	4	44,433,344
*Dpp4*	−0.111	1.13 × 10^−1^	**0.329**	9.32 × 10^−2^	**3.517**	**2.42 × 10^−2^**	0.028	3.80 × 10^−1^	**1.553**	**3.13 × 10^−11^**	0.017	7.28 × 10^−1^	0.211	5.37 × 10^−2^	2	44,333,455
** *Igfbp6* **	**1.110**	**2.04 × 10^−4^**	0.081	6.67 × 10^−1^	**2.796**	2.01 × 10^−1^	**0.821**	**1.64 × 10^−2^**	**2.720**	**3.41 × 10^−10^**	−0.079	4.68 × 10^−1^	**0.325**	**6.21 × 10^−3^**	4	43,445,554
*Adk*	−0.161	5.78 × 10^−2^	**0.338**	8.46 × 10^−2^	**1.000**	**4.86 × 10^−2^**	0.117	1.76 × 10^−1^	**2.226**	**2.74 × 10^−8^**	0.053	6.57 × 10^−1^	0.261	**4.21 × 10^−2^**	3	43,345,555
*Actr10*	−0.120	**1.99 × 10^−2^**	**0.554**	**4.28 × 10^−3^**	**0.816**	7.05 × 10^−2^	0.170	**2.88 × 10^−3^**	−0.207	**4.49 × 10^−3^**	0.129	2.78 × 10^−1^	0.062	5.24 × 10^−1^	0	43,332,233
*Drd1*	**1.070**	**3.46 × 10^−3^**	**−0.279**	5.07 × 10^−1^	0.000	1.00	**1.734**	**1.68 × 10^−3^**	**1.472**	**1.73 × 10^−8^**	−0.077	4.06 × 10^−1^	0.251	**3.38 × 10^−2^**	4	43,332,223
*Tyro3*	**0.267**	**4.15 × 10^−2^**	**−0.202**	4.39 × 10^−1^	**1.572**	**2.11 × 10^−2^**	**−0.438**	**4.39 × 10^−2^**	**1.818**	**1.16 × 10^−9^**	0.128	2.77 × 10^−1^	−0.131	3.40 × 10^−1^	2	43,321,112
*Sox11*	**0.614**	**6.75 × 10^−3^**	**1.267**	**1.63 × 10^−3^**	**1.617**	5.52 × 10^−2^	**1.215**	**9.06 × 10^−6^**	**−0.476**	**1.15 × 10^−4^**	−0.033	7.03 × 10^−1^	−0.205	1.32 × 10^−1^	2	43,321,111
*Trpm3*	**−0.437**	**1.49 × 10^−4^**	0.107	7.64 × 10^−1^	0.000	1.00	**−0.946**	**1.39 × 10^−2^**	**−1.568**	**9.87 × 10^−9^**	−0.188	1.06 × 10^−1^	−0.016	8.63 × 10^−1^	−3	−44,443,334
*Jun*	**−0.439**	**1.30 × 10^−2^**	**−0.609**	3.06 × 10^−1^	**−1.000**	1.07 × 10^−1^	**0.267**	8.55 × 10^−2^	**−2.150**	**7.95 × 10^−11^**	−0.143	2.20 × 10^−1^	−0.003	9.71 × 10^−1^	−2	−44,444,444
*Efnb3*	**−0.383**	**1.12 × 10^−2^**	0.000	1.00	**−0.694**	2.27 × 10^−1^	**−0.686**	**1.56 × 10^−3^**	**−2.080**	**2.00 × 10^−12^**	−0.128	2.74 × 10^−1^	0.197	8.16 × 10^−2^	−3	−44,444,444
*Lct*	**−0.600**	**1.75 × 10^−4^**	**−0.567**	1.15 × 10^−1^	**−1.100**	1.16 × 10^−1^	**0.266**	5.55 × 10^−1^	**−4.916**	**4.72 × 10^−12^**	0.011	3.89 × 10^−1^	**−0.421**	**3.72 × 10^−4^**	−3	−44,444,444
*Pdia6*	**−0.390**	**1.76 × 10^−4^**	**0.440**	2.97 × 10^−1^	**−1.320**	1.05 × 10^−1^	−0.247	**7.76 × 10^−4^**	−0.112	2.60 × 10^−1^	−0.070	5.61 × 10^−1^	**−0.382**	**2.62 × 10^−4^**	−3	−44,444,444
*Kcnq3*	**−0.297**	**9.57 × 10^−3^**	0.000	1.0	0.000	1.00	**−0.452**	**1.47 × 10^−2^**	−0.176	6.74 × 10^−2^	−0.176	1.33 × 10^−1^	**−0.367**	**3.62 × 10^−3^**	−3	−44,444,455
*Mcm6*	**−0.593**	**3.24 × 10^−5^**	**−0.526**	5.41 × 10^−2^	**−1.105**	7.72 × 10^−2^	**−0.419**	4.23 × 10^−1^	**−2.518**	**1.14 × 10^−10^**	−0.028	7.34 × 10^−1^	−0.008	9.38 × 10^−1^	−2	−44,445,666
*Cacna1d*	−0.153	1.65 × 10^−1^	**−0.613**	**1.86 × 10^−2^**	**−0.830**	9.70 × 10^−2^	**−0.309**	**1.29 × 10^−2^**	−0.159	6.13 × 10^−2^	−0.082	4.98 × 10^−1^	−0.240	**4.85 × 10^−2^**	−3	−44,566,677
*Nfia*	**−0.282**	**1.95 × 10^−3^**	0.077	6.30 × 10^−1^	**−0.408**	4.77 × 10^−1^	**−0.373**	**6.99 × 10^−3^**	**−1.022**	**2.60 × 10^−9^**	−0.075	5.19 × 10^−1^	**−0.369**	**5.85 × 10^−3^**	−4	−44,655,556
*Nedd4l*	**−0.276**	**4.08 × 10^−3^**	**−0.653**	**1.53 × 10^−2^**	**−0.283**	3.90 × 10^−1^	**−0.377**	**2.13 × 10^−3^**	**−0.326**	**4.03 × 10^−4^**	−0.079	5.10 × 10^−1^	−0.258	**2.86 × 10^−2^**	−5	−44,677,777
*Fndc1*	**−0.377**	**3.74 × 10^−3^**	0.000	1.00	**−1.356**	1.14 × 10^−1^	**0.361**	1.25 × 10^−1^	**−1.817**	**4.60 × 10^−9^**	−0.113	1.43 × 10^−1^	−0.233	8.75 × 10^−2^	−2	−45,444,444
*Ntf3*	**−0.938**	**2.00 × 10^−3^**	0.208	3.41 × 10^−1^	**−3.919**	2.35 × 10^−1^	**−1.793**	**5.70 × 10^−4^**	**−3.831**	**1.97 × 10^−12^**	−0.156	1.64 × 10^−1^	0.015	8.04 × 10^−1^	−3	−45444,444
** *Itga4* **	**−0.437**	**1.33 × 10^−2^**	−0.007	9.83 × 10^−1^	**−3.476**	6.77 × 10^−2^	**−0.601**	**2.59 × 10^−2^**	**−1.608**	**1.14 × 10^−10^**	−0.012	8.44 × 10^−1^	**−0.490**	**2.28 × 10^−5^**	−4	−45,555,466
*Kbtbd11*	−0.191	**1.58 × 10^−2^**	0.000	1.00	0.000	1.00	**−0.998**	**1.18 × 10^−2^**	**−1.353**	**2.14 × 10^−10^**	−0.158	1.82 × 10^−1^	−0.231	7.09 × 10^−2^	−3	−45,555,555
*Kctd4*	−0.238	**2.69 × 10^−3^**	0.000	1.00	**−0.544**	3.16 × 10^−1^	**−0.516**	**2.34 × 10^−3^**	**−1.775**	**2.59 × 10^−10^**	−0.223	6.47 × 10^−2^	0.021	7.70 × 10^−1^	−3	−45,555,555
** *Tnxb* **	**−0.323**	**4.79 × 10^−4^**	0.000	1.00	**−0.695**	3.40 × 10^−1^	**−0.574**	**7.37 × 10^−3^**	**−1.462**	**6.94 × 10^−9^**	0.000	1.00	**−0.398**	**3.57 × 10^−3^**	−4	−45,555,555
** *Scn3b* **	−0.245	**7.02 × 10^−3^**	**−0.479**	**1.05 × 10^−2^**	**−0.678**	1.15 × 10^−1^	**−0.594**	**5.20 × 10^−5^**	**−0.971**	**4.31 × 10^−8^**	0.178	1.41 × 10^−1^	**−0.304**	**2.13 × 10^−2^**	−5	−45,555,555
*Foxo1*	**−0.500**	**2.14 × 10^−3^**	−0.208	3.19 × 10^−1^	**−0.840**	1.99 × 10^−1^	**−0.587**	**1.09 × 10^−3^**	**−1.836**	**1.47 × 10^−12^**	−0.137	2.26 × 10^−1^	−0.037	7.27 × 10^−1^	−3	−45,566,666
** *Rasgrf1* **	**−0.394**	**5.36 × 10^−5^**	−0.200	2.02 × 10^−1^	**−0.443**	3.55 × 10^−1^	**−0.519**	**4.75 × 10^−4^**	**−1.051**	**2.38 × 10^−10^**	−0.006	9.59 × 10^−1^	**−0.335**	**5.16 × 10^−4^**	−4	−45,666,667
*Auts2*	−0.223	**1.52 × 10^−3^**	0.000	1.00	**−0.491**	3.48 × 10^−1^	**−0.489**	**5.37 × 10^−3^**	**−1.047**	**2.47 × 10^−8^**	−0.117	3.18 × 10^−1^	−0.245	7.51 × 10^−2^	−3	−46,666,666
** *Doc2b* **	**−0.506**	**1.51 × 10^−3^**	**0.371**	1.83 × 10^−1^	**−0.798**	7.63 × 10^−2^	−0.233	**1.07 × 10^−2^**	**−3.232**	**3.42 × 10^−11^**	0.028	7.90 × 10^−1^	**−0.416**	**4.12 × 10^−4^**	−4	−54,444,433
*Dsp*	**−0.749**	**2.12 × 10^−6^**	0.198	4.54 × 10^−1^	**−1.471**	9.31 × 10^−2^	**−1.688**	**9.19 × 10^−3^**	**−5.642**	**5.44 × 10^−14^**	−0.125	1.86 × 10^−1^	−0.023	7.13 × 10^−1^	−3	−55,544,444
*Slc4a4*	**−0.537**	**5.83 × 10^−4^**	0.052	8.62 × 10^−1^	**−1.062**	1.43 × 10^−1^	**−0.912**	**1.93 × 10^−3^**	**−1.025**	**2.36 × 10^−12^**	−0.166	1.61 × 10^−1^	0.043	7.39 × 10^−1^	−3	−55,554,455
** *Pcdh19* **	**−0.572**	**4.43 × 10^−3^**	0.000	1.00	0.000	1.00	−0.252	**1.31 × 10^−2^**	**−1.419**	**9.69 × 10^−15^**	−0.114	3.35 × 10^−1^	**−0.413**	**1.08 × 10^−4^**	−4	−55,555,555
** *Fat4* **	**−0.890**	**4.54 × 10^−4^**	0.000	1.00	0.000	1.00	**−1.352**	**1.80 × 10^−3^**	**−1.748**	**2.48 × 10^−11^**	−0.210	6.03 × 10^−2^	**−0.342**	**1.25 × 10^−2^**	−4	−55,555,555
*Akt3*	−0.224	**5.02 × 10^−3^**	**−0.482**	**1.34 × 10^−2^**	**−0.819**	1.04 × 10^−1^	**−0.372**	**3.84 × 10^−3^**	**−0.602**	**8.66 × 10^−9^**	0.081	4.84 × 10^−1^	−0.128	**1.81 × 10^−1^**	−4	−55,555,555
*Sipa1l2*	**−0.553**	**2.89 × 10^−5^**	0.000	1.00	**−0.804**	1.28 × 10^−1^	**−0.469**	**6.07 × 10^−3^**	**−0.918**	**1.36 × 10^−8^**	−0.200	9.62 × 10^−2^	−0.050	6.56 × 10^−1^	−3	−55,555,556
** *Zfp316* **	−0.218	**4.28 × 10^−3^**	**−1.324**	**3.00 × 10^−4^**	**−1.531**	**1.80 × 10^−2^**	**−0.912**	**8.63 × 10^−3^**	**−0.448**	**1.30 × 10^−3^**	−0.107	3.35 × 10^−1^	0.019	8.72 × 10^−1^	−5	−55,556,666
*Slit1*	**−0.284**	**1.93 × 10^−4^**	−0.198	4.17 × 10^−1^	**−0.807**	1.18 × 10^−1^	0.051	4.40 × 10^−1^	**−2.120**	**6.68 × 10^−12^**	−0.119	3.23 × 10^−1^	**−0.407**	**2.67 × 10^−4^**	−3	−55,566,665
** *Itsn1* **	−0.238	**3.80 × 10^−3^**	**−2.471**	**1.91 × 10^−3^**	**−0.915**	5.91 × 10^−2^	−0.232	**2.93 × 10^−3^**	**−0.785**	**1.52 × 10^−8^**	−0.068	5.71 × 10^−1^	−0.164	1.84 × 10^−1^	−4	−55,777,777

|log_2_FC| > 0.263 and *p*-values ≤ 0.05 are in bold. Consensus score indicates whether FLX induces mainly upregulation (positive score) or downregulation (negative score) of gene expression. Genes in bold are also in Table 6.

**Table 8 ijms-23-13543-t008:** Ontological analysis of downregulated genes (consensus and portrait scores< −2) after FLX treatment in naive animals.

Category	ID	Name	Genes	Count	Fold Enrichment	*Padj*
Cellular component	GO:0030054	cell junction	*Actn2*, *Ahi1*, *Arhgap21*, *Cacna1c*, *Camk2b*, *Ccd85a*, *Chrm1*, *Chrna7*, *Clstn2*, *Ctbp2*, *Ddn*, *Dlg5*, *Dsp*, *Enah*, *Eps8*, *Faim2*, *Flrt3*, *Frmd4b*, *Gabra5*, *Gabrd*, *Gria1*, *Grid1*, *Hap1*, ***Itsn1***, *Kctd12*, *Lgi1*, *Limd1*, *Lrrc7*, *Lrrtm1*, *Map1b*, ***Mpdz***, *Mpp2*, *Nbea*, *Ncdn*, *Nfia*, *Nlgn3*, *Nos1*, *Nphp1*, *Nrxn1*, *Palm*, *Palmd*, *Pcdh1*, *Plpr4*, *Prkcg*, *Prrt1*, *Ptk2b*, *Rapgef2*, *Rgs12*, *Sarm1*, *Scn8a*, *Sema4f*, *Shisa6*, *Slc1a1*, *Sorbs1*, *Sptb*, *Sptbn4*, *Syt7*, *Tenm2*, *Traf4*, *Tspan33*, *Wasf1*	61	2.43	1.70 × 10^−7^
	GO:0045202	synapse	*Ablim3*, *Bcan*, *Cacna1c*, *Calb1*, *Camk2b*, *Capn5*, *Chrm1*, *Chrna7*, *Clstn2*, *Ctbp2*, *Ddn*, *Dlg5*, ***Doc2b***, *Enah*, *Eps8*, *Faim2*, *Gabra5*, *Gabrd*, *Gria1*, *Grid1*, *Grip1*, *Hap1*, *Itsn1*, *Kctd12*, *Lgi1*, *Lrrc7*, *Lrrtm1*, *Lypd1*, ***Map1a***, *Map1b*, ***Mpdz***, *Mpp2*, *Nbea*, *Ncdn*, *Nlgn3*, *Nos1*, *Nrxn1*, *Palm*, *Palmd*, *Plpr4*, *Ppfia4*, *Prkcg*, *Prrt1*, *Rapgef2*, *Rgs12*, *Sarm1*, *Sema4f*, *Shisa6*, *Slc1a1*, *Sorbs1*, *Syt7*, *Tenm2*, *Wasf1*	53	2.23	5.62 × 10^−5^
Biological process	GO:2000310	regulation of NMDA receptor activity	*Dapk1*, *Mapk8ip2*, *Nlgn3*, *Nrxn1*, *Ptk2b*, ***Rasgrf1***, *Rasgrf2*	7	13.9	1.83 × 10^−2^
KEGG pathway	mmu04713	circadian entrainment	*Adcy1*, *Adcy9*, *Adcyap1r1*, *Cacna1c*, *Cacna1d*, *Camk2b*, *Gng7*, *Gria1*, *Nos1*, *Per3*, *Prkcg*, *Ryr1*, *Ryr2*	13	4.82	3.85 × 10^−3^
	mmu04911	insulin secretion	*Abcc8*, *Adcy1*, *Adcy9*, *Adcyap1r1*, *Atf6b*, *Cacna1c*, *Cacna1d*, *Camk2b*, *Gck*, *Prkcg*, *Ryr2*	11	4.65	3.12 × 10^−2^

Genes belonging to the core signatures (Table 6 and Table 7) are in bold.

**Table 9 ijms-23-13543-t009:** Ontological analysis of upregulated genes (consensus and portrait scores ≥4) after FLX treatment.

Category	ID	Name	Genes	Count	Fold Enrichment	*Padj*
Cellular component	GO:0030424	axon	*Bdnf*, *Cdk5r1*, *Cst3*, *Dclk1*, *Dcx*, *Drd1*, *Ntrk2*, *Pink1*, *Ptpn5*, *Sema3a*, *Stx1b*, *Stxbp1*, *Tubb3*	13	4.97	2.26 × 10^−3^
	GO:0009986	cell surface	*Bgn*, *Cd14*, *Cd151*, ***Drd1***, *Ephb6*, *Fcgr4*, *H2-K1*, *H2-Q10*, *Ifitm3*, *Itgav*, *Ntrk2*, *Prlr*, *Tfrc*, ***Tyro3***	14	3.48	3.44 × 10^−2^
	GO:0005576	extracellular region	*Bdnf*, *Bgn*, *Brinp2*, *C1qb*, *Ccdc3*, *Cd14*, *Ccn4*, *Cst3*, *Ephb6*, *Gsn*, ***Igfbp6***, *Lyzl4*, ***Nptx2***, *Olfm3*, *Pcsk2*, *Sema3a*, *Sema3c*, *Serpina3n*, *Spock3*, *Tfrc*, *Vgf*, *Vip*, *Wnt10a*	23	2.37	3.62 × 10^−2^
KEGG pathway	mmu04010	MAPK signaling pathway	*Bdnf*, *Cacna2d2*, *Cd14*, *Gadd45a*, *Kras*, *Mapk9*, *Mapk11*, *Ntrk2*, *Ptpn5*, ***Rasgrp1***	10	5.65	9.09 × 10^−3^
Molecular function	GO:0016301	kinase activity	*Cdk5r1*, *Cdkl5*, *Cmpk1*, *Dclk1*, *Mapk4*, *Mapk9*, *Mapk11*, *Melk*, *Ntrk2*, *Pak5*, *Pink1*, ***Prkar1b***, ***Tyro3***	13	3.73	4.60 × 10^−2^

Genes belonging to the core signature (Appendix A: genes with absolute scores ≥6, Figure 3) are in bold.

**Table 10 ijms-23-13543-t010:** Ontological analysis of downregulated genes (consensus and portrait scores ≤ −4) after FLX treatment.

Category	ID	Name	Genes	Count	Fold Enrichment	*Padj*
KEGG pathway	mmu04010	MAPK signaling pathway	***Akt3***, *Cacna1d*, *Flt3*, *Hspa2*, *Il1r1*, *Map3k4*, *Ntf3*, ***Rasgrf1***, *Rps6kas*, *Stk4*, *Tgfb2*	11	5.17	7.67 × 10^−3^

Genes belonging to the core signature (Appendix A: genes with absolute scores ≥6, Figure 3) are in bold.

## Data Availability

The source code for all the analyses is available at https://github.com/guillaumecharbonnier/mw-ibrahim2022 and a compiled version of the Bookdown report is available at https://guillaumecharbonnier.github.io/mw-ibrahim2022.

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
