# Peer review of "Transcriptomic Studies of Antidepressant Action in Rodent Models of Depression: A First Meta-Analysis"

_ijms, 2022, doi:10.3390/ijms232113543_

Round 1

Reviewer 1 Report

The present work presented by El Chérif Ibrahim et al., reviews transcriptomic alterations exerted by antidepressants (ADs) on models of depression achieved in rodents subjected to stress.

Although the authors found a heterogeneity and variability in different aspects: in animal model, species, sex, AD used, assay methods, body region; they mainly focused on the actions of fluoxetine in hippocampus on patterns of gene expression. The most relevant transcripts they found were immediate early genes and genes related with synaptic plasticity pathways.

Although Major Depression Disorder (MDD) is a very complex condition, the subject is interesting and highly relevant because MDD is increasing worldwide. For this reason, it is important to deeply know the mechanisms and sites of action of ADs from several points of view to decipher their effects.

The work is well documented and clear presented.

Minor comments:

Lines 436 and 439, as well as in the Discussion section: regarding “mechanism of action of AD” and “its selectivity”, it will be beneficial to including other cellular responses in addition to changes in the gene expression: changes in synaptic transmission, direct interaction of ADs with transmitter receptors and ion channels.

In the Discussion section, it would be beneficial for the reader to mention and discuss some aspects related with side- and adverse-effects of ADs, principally on possible consequences of transcriptomic alterations.

Reviewer 2 Report

The authors present a meta-analysis of transcriptomic responses to antidepressants. The study tackles an important issue and is sound from methodological point of view. The results will be useful for scientific community. Therefore, I am convinced that the paper is worth publishing.